# WHISFUSION: PARALLEL ASR DECODING VIA A DIFFUSION TRANSFORMER

## ABSTRACT

Fast automatic speech recognition (ASR) is crucial for applications such as captioning and transcription. Although modern ASR encoders can process up to 30 seconds of audio in a single pass, Whisper-style autoregressive (AR) decoders still generate tokens sequentially, making decoding latency grow linearly with utterance length. We propose Whisfusion, a non-autoregressive (NAR) ASR framework that fuses a frozen pre-trained Whisper encoder with a masked-diffusion text decoder. At each diffusion step, the decoder conditions on the full acoustic context and updates all tokens in parallel, mitigating the AR latency bottleneck while preserving Whisper-compatible generative structure. A lightweight cross-attention adapter trained via parameter-efficient fine-tuning bridges audio and text, and we introduce Parallel Diffusion Decoding (PDD), an ASR-tailored batch-parallel sampling scheme that improves the accuracy–latency trade-off in low-to-mid batch regimes. With 6.5k hours of training data, Whisfusion reaches 4.9% WER on LibriSpeech test-clean, comparable to similarly sized Whisper model (Whisper-small at 5.0%), while enabling much faster decoding. In particular, on 20–30s segments within Whisper's 30s window, Whisfusion reduces decoding time from 674.7 ms to 80.7 ms (8.4× faster) at similar accuracy, demonstrating an efficient NAR operating point for Whisper-compatible ASR.

## 1 INTRODUCTION

### 1.1 THE CHALLENGE OF AUTOREGRESSIVE ASR MODELS

Transformer-based autoregressive models (AR) Vaswani et al. (2017) have achieved state-of-the-art (SOTA) performance in automatic speech recognition (ASR) Dong et al. (2018), with models such as Whisper demonstrating remarkable accuracy across benchmarks. Whisper-small, for example, reports Word Error Rates (WER) of 5.0% and 12.2% on LibriSpeech test-clean/other, setting a strong baseline for open-domain ASR. Extensions like the Two-Pass U2 framework Wu et al. (2021) adapt Whisper for streaming, reducing latency through architectural modifications Yao et al. (2021); Zhou et al. (2025). Yet, sequential token generation inevitably introduces inference latency, limiting effectiveness in real-time ASR Zhou et al. (2025). Managing long-range

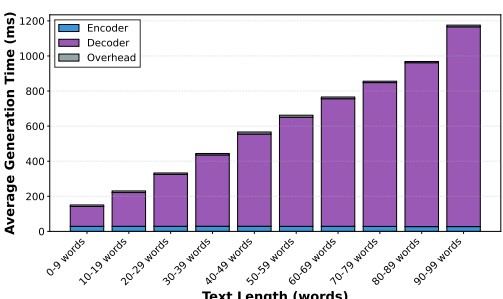

Figure 1: Whisper's processing time scales linearly with text length due to its autoregressive decoder, while encoder time remains constant.

dependencies also adds engineering burden Battenberg et al. (2025), especially in transcription services and on-device ASR. In these environments, sequential decoding causes delays, degrades user experience, and strains computational budgets without a high-performance GPU. The decoder remains the primary bottleneck, a trend illustrated for Whisper-small in Figure 1. Even distilled models designed to mitigate this, such as Whisper-Large-v3-turbo Gandhi et al. (2023), still show a rising decoder time ratio with input length. These challenges highlight the need for alternative decoding paradigms that maintain linguistic coherence while enabling faster, parallelizable inference.

## 1.2 A New Paradigm: Non-Autoregressive Diffusion Transformer Models

Recent work on masked diffusion models (MDMs) Austin et al. (2021); Lou et al. (2024); Shi et al. (2024) has emerged as a promising non-autoregressive alternative for language generation. In contrast to token-by-token generation in AR models, MDMs perform iterative denoising over masked sequences, enabling parallel prediction of multiple tokens at each step. This allows for significantly faster inference while preserving high generation quality. A recent study on the scalability of MDMs Nie et al. (2025) has shown that such models can scale effectively, following power-law scaling laws comparable to AR models under equivalent computing budgets. Since MDM decoding latency is largely independent on output length, it has the potential to overcome the length-scaled latency of current ASR methods that rely on autoregressive transcription.

## 1.3 Our Contribution: Whisfusion

SOTA ASR models suffer from a core architectural mismatch: while their AR decoders are provided with the full acoustic context from a 30-second audio segment, they are structurally limited to processing it sequentially, token-by-token. This inefficient exploitation of the available context creates a significant latency bottleneck. To resolve this trade-off, we propose Whisfusion, a novel non-autoregressive (NAR) framework that fuses a pre-trained Whisper encoder with a text Diffusion decoder. Our main contributions are threefold:

1. **A Novel NAR Framework.** We are the first to propose an architecture that fuses a pre-trained Whisper encoder with a text diffusion decoder for ASR. This novel hybrid NAR framework, enabled by a lightweight PEFT-trained adapter, resolves the context-utilization paradox by allowing the decoder to leverage the full acoustic context in a parallel, non-sequential manner.

2. **A Unique Parallel Decoding Strategy.** We introduce a novel batch-parallel, multi-step decoding strategy that combines random token sampling with a confidence-based refinement mechanism. A key advantage of this approach is the ability to improve accuracy by increasing the number of parallel candidates with negligible impact on inference speed.

3. **Superior Speed-Accuracy Trade-off.** We empirically demonstrate that Whisfusion establishes a new, highly efficient operating point on the speed-accuracy spectrum. Fine-tuned on only 960 hours of LibriSpeech, it is more accurate than Whisper-tiny (8.3% vs. 9.7% WER) while being up to 2.6 times faster on long-form audio. This is driven by its parallel decoder, which achieves a throughput of over 3100 tokens/s—more than 13 times faster than its AR counterpart.

## 2 Background: Text Generation with Diffusion Models

Diffusion models have gained attention for their ability to model complex data distributions through iterative denoising processes. Initially developed for image generation tasks (Ho et al., 2020), these models have been extended to discrete data domains including natural language (Austin et al., 2021). In discrete diffusion models, the forward process typically replaces tokens with a special mask token following a predefined corruption schedule, with more noise gradually added to the data. The reverse process learns to recover the original sequence through a series of denoising steps (Ho et al., 2020).

Compared to autoregressive generation, diffusion-based models offer several advantages, including parallel decoding, bidirectional context modeling, and flexible control over generation dynamics. Nie et al. (2025b) recently introduced LLaDA, an MDM that leverages these advantages to surpass AR baselines in generation speed and to excel at in-context learning, instruction following, and bidirectional reasoning. LLaDA operates by sampling a continuous masking ratio $t \in (0, 1)$, masking each token independently with probability $t$, and training a *mask predictor* $p_\theta(\cdot \mid x_t)$ to infer the original tokens. Its training objective is the expected cross-entropy on masked positions:

$$\mathcal{L}(\theta) \triangleq -\mathbb{E}_{t,x_0,x_t}\Big[\frac{1}{t}\sum_{i=1}^{L}\mathbf{1}\big[x_t^i = \mathrm{M}\big]\,\log p_\theta\big(x_0^i \mid x_t\big)\Big], \tag{1}$$

where the scaling factor $1/t$ equalizes the contribution of examples with different masking levels.

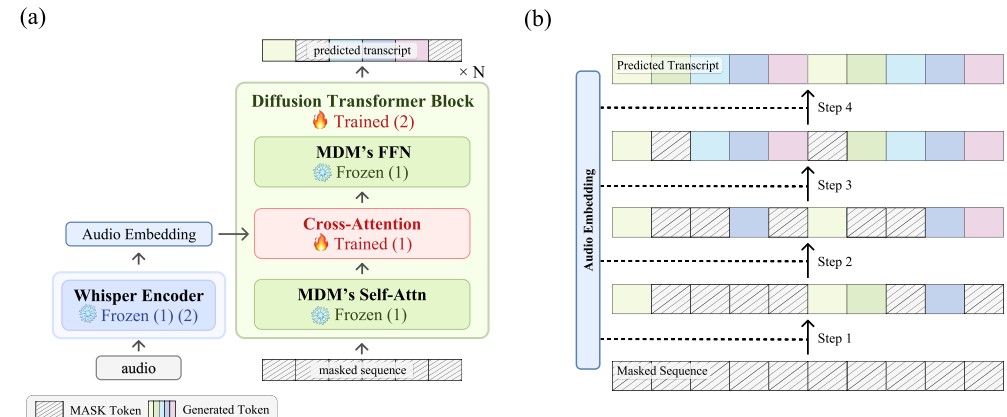

Figure 2: (a) Whisfusion architecture (2-stage training). (b) Decoding process of Whisfusion.

# 3 PROPOSED METHOD: WHISFUSION

In this section, we introduce **Whisfusion**, a novel framework for ASR built upon a Diffusion Transformer. By leveraging the parallel and iterative nature of diffusion models, Whisfusion operates as a NAR system designed for high-speed inference (Figure 2 b). We first present the overall model architecture, which efficiently fuses a pre-trained speech encoder with a text diffusion decoder. We then describe our multi-stage curriculum training strategy designed to achieve both robustness and precision. Finally, we detail our advanced decoding strategy, **Parallel Diffusion Decoding (PDD)**, which overcomes the limitations of conventional NAR decoding by leveraging the unique parallel nature of our model. The detailed architecture of Whisfusion is shown in Table 1.

Table 1: Detailed architecture breakdown of Whisfusion compared to Whisper-Small.

|  | Whisper-Small | Whisfusion |
|---|---|---|
| *Encoder* | 88.2M (shared, frozen) | |
| *Decoder* | | |
| Type | Autoregressive | Diffusion |
| Layers | 12 | 18 |
| Hidden Size | 768 | 768 |
| Parameters | 153.6M | 212.5M |
| (self-attn + cross-attn) | (125.2M + 28.4M) | (170M + 42.5M) |
| **Total Parameters** | 241.7M | 300.7M |
| **Adapter Parameters** | – | 42.5M (9.3%) |

## 3.1 MODEL ARCHITECTURE

The core of Whisfusion is the *fusion* of two pre-trained models from distinct modalities: a speech encoder and a text diffusion decoder. To bridge the gap between Whisper's acoustic representations (audio-to-tensor) and MDM's text-based domain (text-to-text), we insert a lightweight Cross-Attention fusion layer within each block of the MDM's Transformer architecture. Trained via PEFT, this design leverages large pre-trained models while minimizing training costs.

**Speech Encoder:** We utilize the official pre-trained Whisper-small encoder. Trained on 680K hours of diverse audio, it converts raw waveforms into rich high-level acoustic representations (hidden states), providing a robust and generalizable foundation. In the initial training stage, this component remains frozen to preserve its generalized knowledge.

**Diffusion Decoder:** We employ a pre-trained SMDM-170M, a text diffusion transformer, as our decoder. Its inherent non-autoregressive nature allows it to process the entire text sequence in parallel, making it an ideal candidate for high-speed inference. It learns to restore a fully masked text sequence by iteratively denoising it over multiple steps.

**Cross-Attention Fusion Layer:** To enable the text-based MDM decoder to understand the acoustic conditions from the Whisper encoder, we insert a lightweight Cross-Attention layer within each block of the MDM's Transformer architecture. This layer acts as an efficient bridge, enabling each text token to attend to all speech tokens across every decoding step, thereby integrating acoustic context throughout the generation process. This is the only component trained during the initial fine-tuning stage.

## 3.2 Training Strategy: A 2-Stage Curriculum

To effectively train our composite model without catastrophic forgetting, we devise a multi-stage curriculum designed to first establish a robust foundation and then refine the model's performance for the specific challenges of our NAR task (Figure 2 a). We first train only a lightweight adapter with the pre-trained components, then proceed to unfreeze all parameters of the decoder to specialize in our ASR task. Such an adapter-first approach has been shown to mitigate catastrophic forgetting and improve generalization in adapter-based NLP and ASR fine-tuning. Eeckt & hamme (2023); Liu et al. (2024)

**Stage 1: Robust Adapter Training.** Our primary objective in this stage is to teach the Cross-Attention layers to effectively interpret Whisper's acoustic representations and guide the MDM decoder, while preserving the powerful prior knowledge of both base models. To achieve this, we freeze all parameters of both the Whisper encoder and the MDM decoder. Only the newly inserted Cross-Attention layers are trainable. We use the full LibriSpeech 960h dataset (comprising both clean and noisy subsets, train-clean-100 / 360 and train-other-500) to expose the adapter to a wide variety of acoustic conditions, thereby maximizing its robustness and generalization capabilities.

**Stage 2: Full Decoder Harmonization & Specialization.** This stage aims to simultaneously harmonize the pre-trained MDM decoder with the speech-conditioned adapter and specialize the model for the most challenging inference scenario: generating text from a fully masked state. Building upon the Stage 1 model, we unfreeze all parameters of the MDM decoder and fine-tune it jointly with the Cross-Attention adapter. To preserve the hierarchical knowledge within the pre-trained decoder Kenneweg et al. (2022); Awasthi et al. (2022), we apply a layer-wise learning rate decay, where shallower layers are trained with a higher learning rate while deeper, more foundational layers are updated with a smaller learning rate. Critically, this entire stage is conducted exclusively on data samples with a high masking ratio (e.g., 70-100%). This dual-purpose approach forces the decoder's self-attention and feed-forward networks to adapt to the acoustic context while simultaneously becoming experts at generating initial tokens from minimal textual information, thus directly addressing the initial generation stability problem.

## 3.3 Advanced Decoding Strategy: Parallel Diffusion Decoding (PDD)

Standard iterative decoding for NAR models suffers from error propagation, especially when a token is predicted incorrectly with high confidence. Furthermore, popular AR decoding techniques like Beam Search are structurally inefficient for diffusion-style models due to their parallel and fixed-length nature. We therefore propose Parallel Diffusion Decoding (PDD), a novel inference strategy that leverages the unique characteristics of our NAR architecture to efficiently explore multiple candidate transcriptions and select the most probable one.

**Contrasting AR Beam Search and PDD**

In AR models, generating a sequence of length $T$ requires $T$ serial steps, since the $t$-th token cannot be produced independently but depends on the previously generated $t-1$ tokens. This strict sequentiality makes decoding inherently slow and limits throughput even when substantial parallel computation is used within each step. Moreover, beam search produces hypotheses of varying length at intermediate steps, which complicates batching and

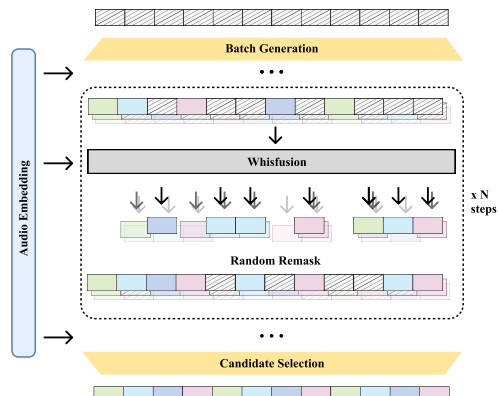

Figure 3: Parallel Diffusion Decoding (PDD) inference. At each of $N$ steps, $k$ candidates are refined in parallel from audio embeddings, iteratively remasked, then selected.

introduces substantial padding overhead and wasted compute. By contrast, Whisfusion conditions on the entire sequence at once, so $k$ hypotheses can be grouped into a single batch and refined simultaneously in each forward pass, reducing redundancy in computation and yielding consistently higher throughput.

**The PDD Algorithm.** Our proposed PDD method (Figure 3), consists of the following steps:

1. **Batch Generation:** At the first step ($t = 1$), instead of selecting a single argmax prediction, we run the decoder $k$ times from the initial token distribution, yielding $k$ diverse candidate sequences $\{y_1^{(1)}, \ldots, y_1^{(k)}\}$ in one forward pass.

2. **Parallel Refinement:** For the subsequent $N-1$ refinement steps ($t = 2, \ldots, N$), we treat these $k$ drafts as a batch. At each step $t$ we *randomly mask* a fixed fraction $\rho_t$ of tokens in every candidate (e.g., $\rho = \{1.0, 0.9, 0.85, 0.80\}$ for $N = 4$) and let the model re-predict the masked positions of each candidate sequences in parallel on a single GPU.

3. **Candidate Selection:** After the final step, we score each of the $k$ complete sequences (e.g., using their average token confidence) and select the highest-scoring sequence as the final output.

This PDD approach minimizes the speed loss typically associated with exploring multiple hypotheses while significantly improving resilience to initial prediction errors, thereby enhancing the final transcription accuracy.

## 4 EXPERIMENTS

To evaluate the effectiveness of our proposed Whisfusion model (301M parameters), we conduct experiments on the widely-used LibriSpeech Panayotov et al. (2015) benchmark, assessing both transcription accuracy and inference speed. We compare Whisfusion against three Whisper variants: Whisper-tiny (39M), the fastest baseline; Whisper-small (244M), which is most comparable in size; and Whisper-large-v3-turbo (809M), a recent model optimized for AR decoding speed (hereafter Whisper-turbo). For latency evaluation, we run each audio file 5 times and report the average to mitigate measurement noise. All evaluation scripts and hyperparameter settings are publicly available for full reproducibility (see Appendix C).

Table 2: Dataset statistics for LibriSpeech train-960h.

| Duration distribution | File count |
|---|---|
| 0–10 seconds | 64,181 (22.8%) |
| 10–20 seconds | 217,005 (77.2%) |
| 20–30 seconds | 55 (<0.1%) |
| **Token statistics** | **Length** |
| 99th percentile | 124 tokens |
| Maximum | 228 tokens |

### 4.1 DATASETS

All experiments are conducted on LibriSpeech, using train-960h for training, dev-clean/other for validation, and test-clean/other for evaluation. Based on the token statistics in Table 2, we set `max_length`=256 for both training and inference, ensuring full coverage of the training data.

### 4.2 IMPLEMENTATION DETAILS

**Base Models and Environment.** Our Whisfusion architecture is built upon two powerful pretrained models: the official openai/whisper-small model as the speech encoder, and the mdm-170M checkpoint from the SMDM project Nie et al. (2025) as the text diffusion decoder. All models were trained and evaluated using 4 x NVIDIA A100 GPUs.

**Stage 1: Adapter Fine-tuning.** The primary goal of this stage was to train the Cross-Attention adapter on the full 960-hour LibriSpeech dataset. For each training sample, we applied a masking ratio chosen uniformly at random from 0% to 100%. This strategy ensures the model is robust across all levels of text corruption. The validation loss converged to a best of 0.0840 (PPL $\approx$ 1.09), indicating the adapter had effectively learned to interpret the acoustic features from the speech encoder.

**Stage 2: Full Decoder Harmonization & Specialization.** Building on the best adapter from Stage 1, this stage unfreezes the MDM decoder and specializes it for initial generation from a fully masked state. To preserve the hierarchical knowledge within the pre-trained decoder, we applied layer-wise learning rate decay. Critically, training was conducted exclusively with a high masking ratio (70-100%). Despite this challenging setting, the model achieved a best validation loss of 0.0958 (PPL $\approx$ 1.10).

# 5 RESULTS AND ANALYSIS

## 5.1 MAIN RESULTS

We present the primary quantitative results of Whisfusion on the LibriSpeech benchmark in Table 4 and Table 3. All experiments use k = 15 candidates and N = 4 refinement steps unless otherwise stated. The masking ratios for the four steps are 1.0, 0.9, 0.85 and 0.8, respectively.

On the test-clean set, Whisfusion achieves a WER of 8.3%, representing a 14% relative improvement over Whisper-tiny (9.7% WER). The RTF measurements show that Whisfusion (0.0165) outperforms Whisper-tiny (0.0176), while being 2.4× faster than Whisper-small (0.0397) and 2.3× faster than Whisper-turbo (0.0374). On the more challenging test-other set, Whisfusion maintains competitive performance with 17.0% WER, positioning itself between Whisper-tiny and Whisper-small in terms of accuracy.

Table 3 reveals the distinct characteristics of our non-autoregressive architecture across different audio durations. While autoregressive models show varying inference times dependent on sequence length, Whisfusion maintains nearly constant total inference time: 122.3ms for 0-10s audio, 123.1ms for 10-20s, and 120.1ms for 20-30s segments. This consistency translates to dramatic RTF improvements as audio length increases—from 0.029 for short segments to 0.005 for longer ones, a 5.80× improvement. In contrast, Whisper models show more modest scaling: Whisper-tiny improves only 1.57× (0.022 to 0.014), while Whisper-small and Whisper-turbo show similar limited gains.

Notably, while Whisfusion demonstrates strong performance on audio segments up to 20 seconds, we observe degraded accuracy on the 20-30s category (15.9% WER). This degradation can be attributed to the severe scarcity of long-form audio in the training data: among 281,241 training samples in LibriSpeech train-960h, only 55 files (0.02%) exceed 20 seconds (see Table 2), so the model struggles to generalize for such sentences.

The decoder performance metrics highlight the fundamental difference between autoregressive and non-autoregressive approaches. Whisfusion achieves a throughput of over 3,180 tokens per second with a consistent 0.31 ms per token across all duration categories. This represents a 16× improvement over Whisper-tiny (190-240 tokens/s) and 36× over Whisper-small (83-103 tokens/s). Furthermore, while the decoder component dominates inference time in Whisper models—accounting for 80-95% of total computation as sequences lengthen—it remains fixed at approximately 67% for Whisfusion regardless of audio duration.

The time breakdown analysis shows that Whisfusion allocates 23-24% of computation to the encoder, compared to 3-8% for Whisper-tiny and 6-14% for Whisper-small. This reallocation is enabled by the efficiency of parallel decoding, which completes in a fixed 82ms regardless of sequence length, while autoregressive decoders scale from 82ms to 292ms (Whisper-tiny) or 187ms to 675ms (Whisper-small) as audio duration increases from 0-10s to 20-30s.

Table 3: Performance across durations on LibriSpeech *test-clean*.

| Duration | Model | Acc. (%) | | Time (ms) | | | E2E speed | | | Decoder |
|---|---|---|---|---|---|---|---|---|---|---|
| | | WER | CER | Enc | Dec | Ovhd | Total ↓ | RTF ↓ | Speed ↑ | tok/s ↑ |
| 0–10 s | Whisper-tiny | 10.5 | 4.5 | 7.6 | 82.1 | 12.7 | 102.4 | 0.022 | 2.18× | 190.3 |
| | Whisper-small | 5.4 | 2.3 | 32.2 | 187.3 | 8.8 | 228.1 | 0.048 | 1.00× | 83.3 |
| | Whisper-turbo | 3.8 | 1.5 | 156.3 | 85.8 | 8.6 | 250.6 | 0.057 | 0.84× | 177.7 |
| | **Whisfusion** | **7.9** | **2.7** | **29.1** | **82.1** | **11.1** | **122.3** | **0.029** | **1.66×** | **3186.1** |
| 10–20 s | Whisper-tiny | 7.0 | 2.6 | 7.2 | 187.1 | 11.6 | 205.9 | 0.015 | 2.33× | 230.3 |
| | Whisper-small | 3.5 | 1.2 | 30.3 | 435.5 | 9.1 | 475.0 | 0.035 | 1.00× | 99.6 |
| | Whisper-turbo | 2.5 | 0.7 | 155.7 | 184.5 | 9.1 | 349.1 | 0.026 | 1.35× | 218.1 |
| | **Whisfusion** | **8.0** | **2.6** | **29.4** | **82.0** | **11.6** | **123.1** | **0.009** | **3.89×** | **3183.7** |
| 20–30 s | Whisper-tiny | 6.4 | 2.4 | 7.4 | 292.3 | 14.2 | 313.9 | 0.014 | 2.21× | 238.8 |
| | Whisper-small | 3.7 | 1.2 | 29.0 | 674.7 | 9.9 | 713.5 | 0.031 | 1.00× | 102.9 |
| | Whisper-turbo | 2.6 | 0.7 | 155.6 | 285.6 | 8.6 | 449.9 | 0.020 | 1.55× | 230.0 |
| | **Whisfusion** | **15.9** | **7.7** | **29.0** | **80.7** | **10.3** | **120.1** | **0.005** | **6.20×** | **3188.6** |

Table 4: WER and CER on LibriSpeech test sets (clean/other) for Whisper variants vs. Whisfusion.

| Model | test-clean | | test-other | |
|---|---|---|---|---|
| | WER (%) ↓ | CER (%) ↓ | WER (%) ↓ | CER (%) ↓ |
| Whisper-tiny | 9.7 | 4.1 | 22.5 | 11.8 |
| Whisper-small | 5.0 | 2.1 | 12.2 | 6.2 |
| Whisper-turbo | 3.5 | 1.4 | 6.6 | 2.8 |
| Whisfusion | 8.3 | 2.9 | 17.0 | 6.9 |

## 5.2 ABLATION STUDIES

To validate the effectiveness of each component in Whisfusion, we conduct comprehensive ablation studies on the LibriSpeech test-clean dataset. The results demonstrate the importance of our key design choices in achieving the final performance.

### 5.2.1 IMPACT OF 2-STAGE TRAINING STRATEGY

Table 5 demonstrates the critical importance of our design choices. The "w/o Acoustic Conditioning" experiment, where we remove the cross-attention adapter, confirms the model's heavy reliance on acoustic information. Despite masking only 30% of the tokens from the ground truth transcript, the model produced near-random transcriptions with a WER of 150.8%, indicating that it fails to generate meaningful outputs without acoustic guidance. Furthermore, the results validate our 2-stage curriculum. The Stage 1 model provides a strong foundation (10.3% WER), which is improved to 9.0% after the initial Stage 2 fine-tuning. Crucially, the final specialization on high-mask-ratio samples is what enables the model to achieve its optimal performance of 8.3% WER.

Table 5: Each component, from acoustic conditioning to the 2-stage curriculum, contributes to the final performance.

| Model configuration | WER (%) ↓ |
|---|---|
| Whisfusion (Full model) | **8.3** |
| *Acoustic conditioning* | |
| w/o acoustic conditioning | 150.8 |
| *Training strategy* | |
| Stage 1 only | 10.3 |
| w/o high-ratio fine-tuning | 9.0 |

### 5.2.2 IMPACT OF PARALLEL DIFFUSION DECODING (PDD)

Table 6 assesses the effectiveness of our PDD strategy. A unique characteristic of our approach is that, due to its batch-parallel nature, increasing the number of candidates (k) has a minimal impact on inference speed, with the primary cost being memory consumption. As shown in the table, increasing k from 5 to 15 progressively lowers the WER from 9.1% to 8.3%, while the RTF remains remarkably stable around 0.017-0.021.

This profile offers a significant advantage over single-sequence decoding. For instance, PDD with k=15 achieves a much lower WER than the fast 4-step single-sequence baseline (8.3% vs. 12.8%) at a comparable RTF. It is also significantly faster than the 15-step single-sequence baseline while being considerably more accurate. Therefore, for our main experiments, we select $k = 15$ to achieve the best accuracy within this highly efficient latency profile. The Oracle WER column further reveals the potential of our generated candidates, suggesting that performance could be improved even more with an advanced selection mechanism.

**PDD Selection Accuracy.** Our confidence-based selection mechanism demonstrates strong performance. As detailed by our analysis, it correctly identifies the best candidate (i.e., the one with the lowest WER) in 68.7% of cases. This results in an average selection gap of only 2.4% WER between our model's actual WER (8.3%) and the oracle WER (5.9%). Furthermore, the selected candidate is near-optimal in the majority of cases, falling within a 2% WER gap of the best possible outcome 69.3% of the time. This high selection accuracy validates the effectiveness of our confidence scoring approach.

Table 6: Comparison of decoding strategies.

| Decoding strategy | WER (%) ↓ | RTF ↓ | Oracle WER (%) |
|---|---|---|---|
| Single sequence (4 steps) | 12.8 | 0.018 | – |
| Single sequence (15 steps) | 10.1 | 0.059 | – |
| PDD ($k$=5, 4 steps) | 9.1 | 0.019 | 7.4 |
| PDD ($k$=10, 4 steps) | 8.7 | 0.021 | 6.5 |
| PDD ($k$=15, 4 steps) | 8.3 | 0.017 | 5.9 |

### 5.2.3 STEP-WISE ANALYSIS

Table 7 reveals the iterative refinement process. The model makes aggressive predictions in early steps (96% token changes), then progressively refines its output. Most dramatic improvements occur in Step 2, where WER drops from 42.3% to 24.6% while only 12% of tokens change—indicating that the model quickly converges to near-final predictions. By Step 3, with only 9% of tokens changing, the model achieves most of its final accuracy (18.9% WER). The final step serves as fine-tuning, modifying just 7% of tokens for a modest improvement to 16.9% WER. The monotonic increase in average confidence (0.77→0.95) strongly correlates with WER reduction, validating our confidence-based selection strategy.

Table 7: Progressive improvement across diffusion steps.

| Step | Mask ratio | WER (%) ↓ | Avg conf. ↑ | Tokens changed ↓ |
|---|---|---|---|---|
| 0 | 100% | – | – | – |
| 1 | 90% | 42.3 | 0.77 | 96% |
| 2 | 85% | 24.6 | 0.90 | 12% |
| 3 | 80% | 18.9 | 0.93 | 9% |
| 4 | 0% | 16.9 | 0.95 | 7% |

### 5.3 QUALITATIVE ANALYSIS

**Visualization of Iterative Refinement.** To illustrate the working mechanism of our diffusion decoder, Figure 4 visualizes how a transcription is gradually refined over multiple decoding steps. The process starts from a fully masked sequence and iteratively corrects and specifies tokens to form a coherent sentence.

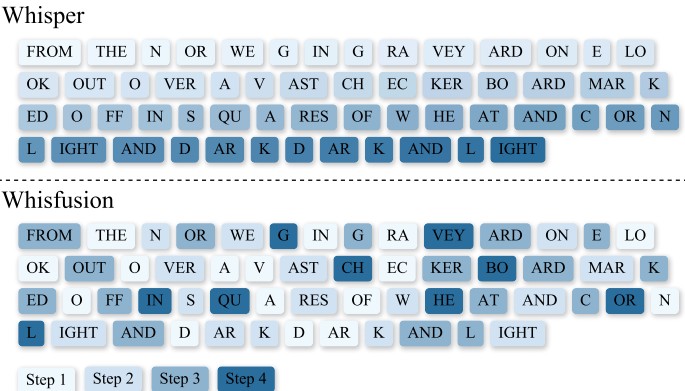

Figure 4: Qualitative comparison of the decoding process. Darker colors indicate tokens finalized in later steps.

# 6 RELATED WORK: AR AND NAR ASR MODELS

Decoding in ASR centers on two paradigms: alignment-based NAR and sequential AR, reflecting the trade-off between efficiency and accuracy.

Early NAR approaches, especially those using Connectionist Temporal Classification (CTC), gained traction for their efficient frame-level parallel inference Graves et al. (2006). CTC maps acoustic frames to tokens by marginalizing over alignments, removing the need for frame-level supervision. However, its assumption of conditional independence limits modeling of long-range dependencies, often leading to incoherent or grammatically flawed transcriptions in noisy or open-domain settings.

To overcome these issues, refinement-based methods like Mask-CTC were proposed Higuchi et al. (2020). Mask-CTC improves initial predictions by masking low-confidence tokens and refining them with a masked language model. While accuracy improves, it inherits CTC's fixed-length constraint, preventing correction of insertion/deletion errors. It also lacks the flexibility for free token generation or reordering.

By contrast, autoregressive models such as Whisper employ a Transformer encoder-decoder that generates tokens sequentially, conditioning each prediction on all prior tokens Radford et al. (2023). This sequential decoding enables rich contextual modeling and has become the standard in high-accuracy, open-domain ASR. Whisper achieves strong results on multilingual and multitask benchmarks, but the sequential nature of AR decoding causes high latency. Even in distilled or optimized variants, the decoder often dominates runtime in long-form transcription.

# 7 CONCLUSION AND FUTURE WORK

## 7.1 CONCLUSION

In this work, we addressed the inherent latency bottleneck of autoregressive ASR models. We introduced Whisfusion, a novel framework that efficiently fuses a pre-trained Whisper encoder with a non-autoregressive text diffusion decoder using a lightweight, parameter-efficient adapter. Our extensive experiments on the LibriSpeech benchmark demonstrate that Whisfusion establishes a new, highly effective operating point on the speed-accuracy spectrum. It achieves a lower WER than Whisper-tiny and showcases a superior scalability profile, becoming significantly faster on long-form audio where traditional AR models falter. Furthermore, we proposed Parallel Diffusion Decoding (PDD), a batch-parallel search strategy that uniquely allows for improving accuracy by increasing the number of parallel candidates with negligible impact on inference speed. Our work validates that diffusion-based decoders are a powerful and viable alternative to conventional AR models, paving the way for high-throughput, low-latency ASR systems.

## 7.2 FUTURE WORK

Several promising avenues exist for future research. The most significant direction is large-scale training. We expect that training the Whisfusion architecture on a large, multilingual dataset, similar to the 680K hours used for the original Whisper, would allow the model to retain Whisper's celebrated robustness and zero-shot capabilities. Such an approach could yield a model that combines the high accuracy of large AR models with the exceptional speed of our NAR framework.

Furthermore, the architectural blueprint of Whisfusion opens possibilities beyond ASR. Its ability to generate diverse hypotheses in parallel with minimal speed trade-off makes it particularly suitable for novel applications. For instance, it could be extended to simultaneous multi-language translation and transcription, where target languages are treated as candidates within the same batch—a task infeasible for AR models. This also makes it suitable for domains where exploring a solution space is critical, such as robotics (e.g., generating multiple action plans) or multi-task learning.

Other future work includes exploring architectural enhancements. For mobile and on-device scenarios, further model compression through techniques like layer dropping or progressive distillation could be investigated. Finally, refining the PDD strategy, perhaps by training a lightweight rescoring model to select candidates, could help close the gap to the Oracle WER and further boost performance.

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

# A APPENDIX

# B ALGORITHMS AND THEORETICAL BASIS

This section provides the technical details of Whisfusion's core components. We present the pseudocode for our two main contributions: the 2-Stage Curriculum Training strategy and the Parallel Diffusion Decoding (PDD) strategy. We also briefly discuss the theoretical foundations of the Masked Diffusion Model that our decoder is based upon. The complete source code for all algorithms and experiments is provided in the Supplementary code for full reproducibility.

## B.1 TRAINING ALGORITHM

Algorithm 1 details the procedure for our 2-stage curriculum training, as described in the main paper. As shown, the key difference between the stages lies in the scope of trainable parameters and the distribution of the masking ratio t. For all training batches, the input text is tokenized and padded to a fixed maximum length of 256, a value chosen based on the token distribution of the training data (Table 2).

---

**Algorithm 1** 2-Stage Curriculum Training for Whisfusion

---

**Require:** Whisper Encoder $\mathcal{E}_\phi$, MDM Decoder $\mathcal{D}_\theta$, Adapter $\mathcal{A}_\psi$, Dataset $\mathcal{D}$
**Ensure:** Trained Whisfusion model $\{\phi, \theta, \psi\}$
 1: **— Stage 1: Robust Adapter Training —**
 2: Freeze encoder parameters $\phi$ and decoder parameters $\theta$
 3: **for** each epoch = 1 to $N_1$ **do**
 4:    **for** each batch $(x_{audio}, y_{text}) \in \mathcal{D}$ **do**
 5:       $C \leftarrow \mathcal{E}_\phi(x_{audio})$ {Extract acoustic features}
 6:       $t \sim \mathcal{U}(0, 1)$ {Sample uniform masking ratio}
 7:       $y_{masked} \leftarrow \text{MASK}(y_{text}, t)$ {Apply masking to text}
 8:       $\hat{y} \leftarrow \mathcal{D}_\theta(y_{masked}, C, \mathcal{A}_\psi)$ {Decode with adapter}
 9:       Compute loss using Eq. 3
10:       Update $\psi \leftarrow \psi - \alpha \nabla_\psi \mathcal{L}$
11:    **end for**
12: **end for**
13: **— Stage 2: Full Decoder Harmonization —**
14: Freeze only encoder parameters $\phi$
15: Initialize layer-wise learning rates: $\alpha_l = \alpha_{base} \cdot \gamma^{(L-l)}$
16: **for** each epoch = 1 to $N_2$ **do**
17:    **for** each batch $(x_{audio}, y_{text}) \in \mathcal{D}$ **do**
18:       $C \leftarrow \mathcal{E}_\phi(x_{audio})$
19:       $t \sim \mathcal{U}(0.7, 1.0)$ {High masking ratio only}
20:       $y_{masked} \leftarrow \text{MASK}(y_{text}, t)$
21:       $\hat{y} \leftarrow \mathcal{D}_\theta(y_{masked}, C, \mathcal{A}_\psi)$
22:       Compute loss using Eq. 3
23:       Update $\{\theta, \psi\}$ with layer-wise learning rates $\{\alpha_l\}$
24:    **end for**
25: **end for**
26: **return** Trained model parameters $\{\phi, \theta, \psi\}$

---

Our training objective is adapted from the standard Masked Diffusion Model (MDM) loss function.

$$\mathcal{L}_{MDM} = -\mathbb{E}_{t,x_0,x_t}\Big[\frac{1}{t}\sum_{i=1}^{L}\mathbf{1}[x_t^i = \mathbf{M}]\,\log p_\theta\big(x_0^i \mid x_t\big)\Big] \qquad (2)$$

As proven by Ou et al. (2025), this loss function serves as an upper bound on the negative log-likelihood of the model distribution ($-\mathbb{E}_{y_0 \sim p_{\text{data}}(y_0)}[\log p_\theta(y_0)] \leq \mathcal{L}$), ensuring that minimizing our objective corresponds to a principled maximum likelihood estimation framework.

For Whisfusion, we adapt this objective to be conditioned on the acoustic features $C = \mathcal{E}_\phi(x_{audio})$ provided by the Whisper encoder. The model must predict the original text tokens $y_0$ given both the masked text $y_t$ and the acoustic condition $C$. The trainable parameters are the decoder weights $\theta$ and the adapter weights $\psi$. Our final loss function is therefore:

$$\mathcal{L}(\theta, \psi) \triangleq -\mathbb{E}_{x_{audio}, y_0, t, y_t}\Big[\frac{1}{t} \sum_{i=1}^{L} \mathbf{1}\big[y_t^i = \mathbf{M}\big] \log p_{\theta, \psi}\big(y_0^i \mid y_t, C\big)\Big] \tag{3}$$

The key insight of our approach is that by conditioning the diffusion process on rich acoustic features from a pre-trained encoder, we can leverage the parallel generation capabilities of diffusion models while maintaining the acoustic fidelity necessary for accurate speech recognition. This formulation allows the model to iteratively refine its predictions based on both the partially observed text sequence and the complete acoustic context, effectively combining the strengths of both autoregressive ASR models (acoustic modeling) and non-autoregressive text generation (parallel decoding).

### B.2 PARALLEL DIFFUSION DECODING (PDD) ALGORITHM

Algorithm 2 formalizes this three-stage process of hypothesis generation, parallel refinement, and final selection. The key architectural advantage of this approach over traditional AR Beam Search is summarized in Table 8. While AR models require a number of sequential steps proportional to the output length (T), PDD completes in a small, fixed number of steps (N), making it fundamentally more scalable for long-form audio.

---

**Algorithm 2** Parallel Diffusion Decoding (PDD)

---

**Require:** Acoustic condition $C$, Model (Whisfusion) $M$
**Require:** Number of candidates $k$, Number of steps $N$
**Ensure:** Best transcription $y^*$
 1: **— 1. Batch Generation —**
 2: $Y_0 \leftarrow$ Initialize a batch of $k$ masked sequences
 3: Logits $\leftarrow M(Y_0, C)$ {Single forward pass for all k}
 4: $Y_1 \leftarrow$ Sample$(k, \text{Logits})$ {Sample k initial hypotheses}
 5: **— 2. Parallel Refinement —**
 6: **for** $t = 1$ to $N - 1$ **do**
 7: $\quad Y_{masked} \leftarrow$ ApplyMaskingStrategy$(Y_t)$
 8: $\quad$ Logits $\leftarrow M(Y_{masked}, C)$
 9: $\quad Y_{t+1} \leftarrow$ UpdateUnmaskedTokens(Logits, $Y_{masked}$)
10: **end for**
11: **— 3. Candidate Selection —**
12: $Y_{final} \leftarrow Y_N$
13: Scores $\leftarrow$ CalculateConfidence$(Y_{final})$
14: $y^* \leftarrow Y_{final}[\text{argmax}(\text{Scores})]$
15: **return** $y^*$

---

Table 8: Comparison of Autoregressive Beam Search and our Parallel Diffusion Decoding (PDD). The key advantage of PDD is its fixed, small number of sequential steps, independent of the output length.

| Aspect | AR Beam Search | PDD (Ours) |
|---|---|---|
| **Sequential Steps** | $T$ (Output Length) | $N$ (Fixed, e.g., 4) |
| **Work per Step** | Batch of $k$ beams | Batch of $k$ full sequences |
| **GPU Parallelism** | High *within* each step | High *within* each step |
| **Primary Bottleneck** | Sequential dependency across $T$ steps | Memory for $k$ candidates |
| **Typical Model Calls** | $T \approx 100 - 200$ | $N = 4$ |

# C  HYPERPARAMETER SETTINGS

This section provides a comprehensive list of the key hyperparameters used for our 2-stage training curriculum to ensure full reproducibility. All training was conducted on 4 x NVIDIA A100 40GB GPUs. Table 9 details the specific settings for the final Whisfusion model.

**Rationale for Stage 1.**  The primary goal of Stage 1 is to robustly train the newly initialized adapter. We use a relatively high learning rate (1e-4) and a large effective batch size (512) to ensure stable and efficient learning on the diverse 960-hour dataset. Training with a uniform masking ratio (0-100%) exposes the adapter to all levels of text corruption, forcing it to learn a generalizable mapping from acoustic features to textual context.

**Rationale for Stage 2.**  The goal of Stage 2 is to fine-tune the entire pre-trained MDM decoder while preserving its powerful learned representations. This requires a more delicate approach. We use a much lower base learning rate (1e-5) to prevent catastrophic forgetting. Critically, we apply layer-wise learning rate decay (LLRD). We empirically observed that fine-tuning the entire decoder with a single learning rate led to training instability and performance collapse. LLRD was therefore a necessary choice to gently update the foundational lower layers while allowing the upper layers to adapt more quickly to the ASR task. Finally, training exclusively on high masking ratios (70-100%) specializes the model for the most challenging part of inference: generating the initial transcript from a fully masked state.

For the ablation study model labeled "w/o high-ratio fine-tuning", the training settings are identical to Stage 2, with the sole exception that the masking ratio was kept at a uniform (0-100%) distribution.

Table 9: Key training hyperparameters for the final Whisfusion model.

| Hyperparameter | Stage 1 (Adapter Training) | Stage 2 (Specialization) |
|---|---|---|
| **Trainable Components** | Adapter Only | Adapter + Decoder |
| *Optimizer & Scheduler* | | |
| Optimizer | AdamW | AdamW |
| Learning Rate (Base) | 1e-4 | 1e-5 |
| LR Scheduler | Cosine (Epoch) | Cosine (Step) |
| Warmup Ratio | 0.02 | 0.1 |
| Layer-wise LR Decay Rate | N/A | 0.9 |
| Weight Decay | 0.01 | 0.005 |
| *Training Configuration* | | |
| Effective Batch Size | 512 | 256 |
| Max Epochs | 80 | 30 |
| Early Stopping Patience | 8 | 5 |
| Masking Ratio | Uniform (0-100%) | Uniform (70-100%) |

# D  TRAINING DYNAMICS

Figure 5 and Figure 6 summarize learning behavior across the two stages. In **Stage 1** (adapter training), the loss drops rapidly then stabilizes; validation closely tracks training with no overfitting, and error rates plateau after early epochs. The train–validation gap narrows as the adapter aligns acoustic and textual representations under wide masking, indicating robust generalization.

In **Stage 2** (decoder fine-tuning), losses start low and decrease smoothly. WER shows small oscillations before settling, while CER stays consistently low, suggesting preserved pre-trained knowledge. Layer-wise learning-rate decay (LLRD) damps deep-layer fluctuations while letting upper layers adapt. Best validation-loss and best-WER epochs do not exactly coincide; early stopping selects stable minima over transient dips. Overall, both stages show steady improvement and stable convergence, supporting the effectiveness of the two-stage strategy.

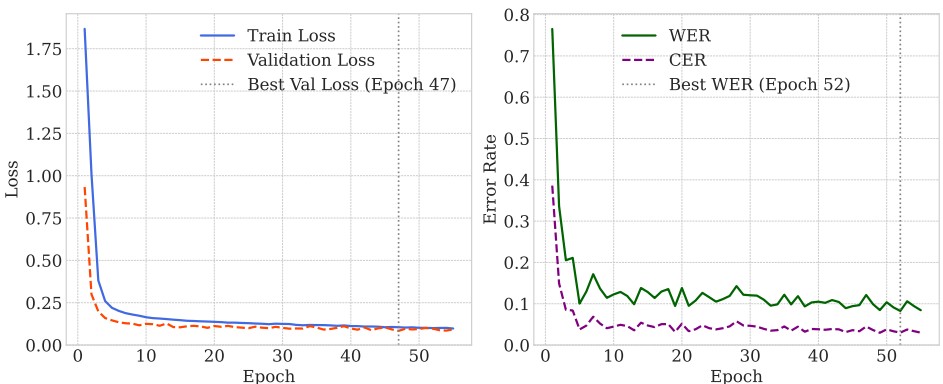

Figure 5: Stage 1 training dynamics (adapter).

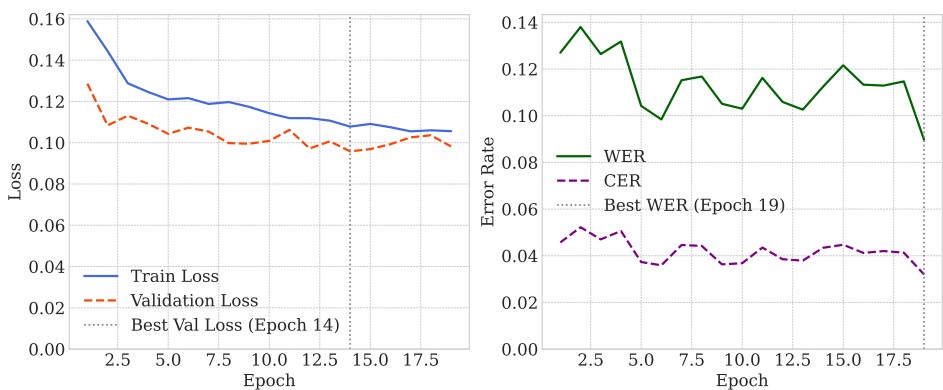

Figure 6: Stage 2 training dynamics (decoder).

# E    IN-DEPTH MODEL ANALYSIS

In this section, we present additional analyses to provide deeper insights into key characteristics of our Whisfusion model: its ability to predict sequence length, the reliability of its confidence scores, and its performance on long utterances.

## E.1    LENGTH ESTIMATION ACCURACY

A key challenge for non-autoregressive models is predicting the correct output length without sequential cues. Figure 7 analyzes Whisfusion's length estimation performance. The scatter plot on the left shows a strong linear correlation between the ground truth and predicted lengths, indicating that our model generally learns to estimate the target sequence length effectively from the acoustic features. However, the plot also reveals increased variance and larger errors for longer sequences. This is consistent with the observation made in the main paper: the model's performance degrades on long-form audio due to the severe scarcity of such examples in the training data (less than 0.1% of the training set is longer than 20 seconds). The plot on the right further confirms that these larger length estimation errors directly correlate with higher WER, highlighting the importance of accurate length prediction for overall performance.

## E.2    CONFIDENCE-ACCURACY CORRELATION

Our Parallel Diffusion Decoding (PDD) strategy relies on confidence scores to select the best candidate. Figure 8 validates this approach by analyzing the relationship between the model's average output confidence and the actual WER for each sample. The scatter plot (left) and the box plot (right)

both demonstrate a clear negative correlation: higher confidence scores consistently correspond to lower error rates. This strong correlation indicates that our model's confidence is well-calibrated and serves as a reliable proxy for transcription accuracy, justifying its use as the selection criterion in PDD.

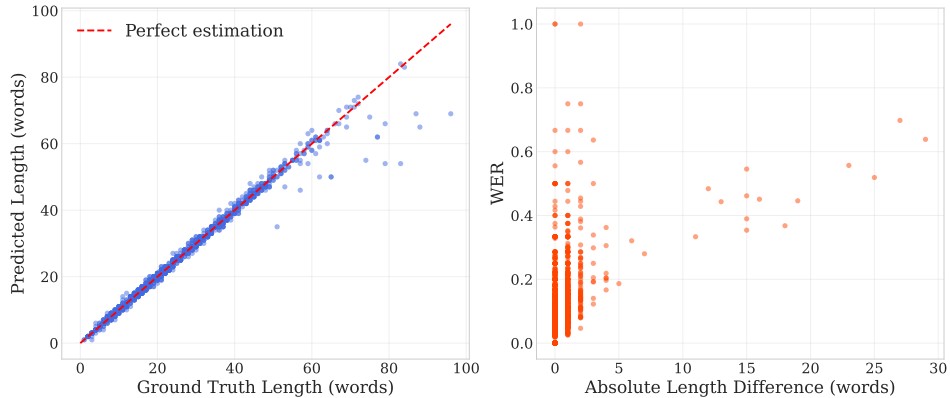

Figure 7: Analysis of Whisfusion's length estimation accuracy. (Left) Predicted length vs. ground truth length. (Right) Absolute length difference vs. WER.

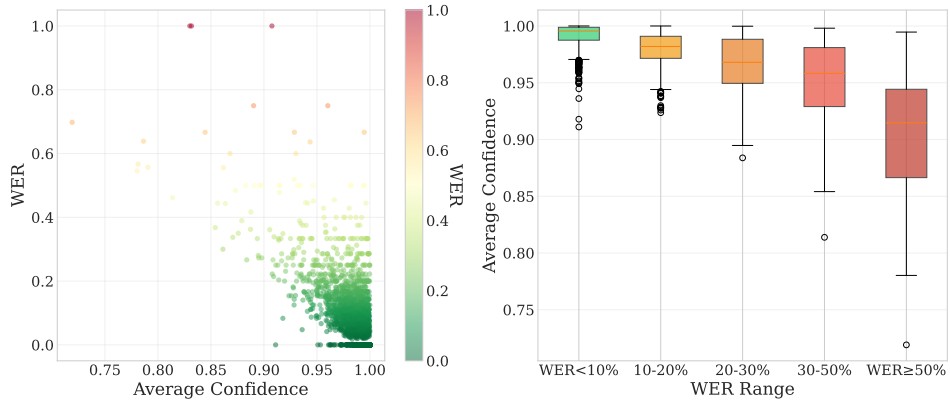

Figure 8: Correlation between average token confidence and WER. (Left) Scatter plot showing a negative correlation. (Right) Box plot showing the distribution of confidence scores for different WER ranges.

## F  ABLATION STUDY ON PDD PARAMETERS

This section details the experiments conducted to determine the optimal values for the number of candidates ($k$), refinement steps ($N$), and the masking schedule in our PDD strategy.

### F.1  IMPACT OF NUMBER OF CANDIDATES ($k$)

First, we examine how the number of parallel candidates affects accuracy while keeping other parameters fixed ($N = 4$, standard masking schedule [1.0, 0.9, 0.85, 0.8]). As shown in Table 10, increasing $k$ from 5 to 15 yields a consistent improvement in both the final selected WER and the potential Oracle WER. This demonstrates the effectiveness of exploring a wider hypothesis space, as a larger pool of candidates increases the probability of finding a more accurate transcription.

Table 10: Effect of number of candidates ($k$) on WER

| Candidates ($k$) | WER (%) | Oracle WER (%) |
|:---:|:---:|:---:|
| 5 | 9.09 | 7.44 |
| 10 | 8.65 | 6.45 |
| 15 | 8.34 | 5.88 |

### F.2 IMPACT OF NUMBER OF STEPS ($N$)

Next, we investigate the effect of varying the number of refinement steps ($N$) while keeping $k = 5$. Table 11 shows that performance improves steadily as $N$ increases. However, we observe diminishing returns beyond 4-6 steps; for example, doubling the steps from 4 to 8 only yields a 0.9% absolute WER reduction. This suggests that a small number of refinement steps is sufficient for the model to converge to a high-quality solution.

Table 11: Effect of number of refinement steps ($N$) on WER.

| Steps ($N$) | WER (%) | Oracle WER (%) |
|:---|:---:|:---:|
| 2 ([1.0, 0.85]) | 14.27 | 12.24 |
| 3 ([1.0, 0.9, 0.8]) | 9.70 | 7.89 |
| 4 ([1.0, 0.9, 0.85, 0.8]) | 9.09 | 7.44 |
| 5 ([1.0, 0.95, 0.9, 0.85, 0.8]) | 8.69 | 6.85 |
| 6 ([1.0, 0.96, 0.92, 0.88, 0.84, 0.8]) | 8.46 | 6.72 |
| 8 ([1.0, ..., 0.65]) | 8.19 | 6.46 |

### F.3 IMPACT OF MASKING SCHEDULE

Finally, we explore different masking schedules with fixed $k = 5$ and $N = 4$. The masking schedule dictates the pace of the denoising process. As shown in Table 12, a standard, gradual decay schedule performs best. While a conservative schedule yields comparable results, aggressive schedules that unmask tokens too quickly (e.g., [1.0, 0.7, 0.5, 0.3]) significantly degrade performance, highlighting the importance of a gradual, iterative refinement process.

Table 12: Effect of different masking strategies on WER.

| Masking Strategy | WER (%) | Oracle WER (%) |
|:---|:---:|:---:|
| *Standard:* | | |
| [1.0, 0.9, 0.85, 0.8] | 9.09 | 7.44 |
| *Conservative (slow decay):* | | |
| [1.0, 0.98, 0.95, 0.9] | 9.48 | 7.54 |
| [1.0, 0.95, 0.9, 0.85] | 9.07 | 7.34 |
| *Aggressive (fast decay):* | | |
| [1.0, 0.85, 0.7, 0.6] | 9.51 | 7.72 |
| [1.0, 0.7, 0.5, 0.3] | 12.90 | 10.35 |

### F.4 SUMMARY AND CONFIGURATION CHOICE

Based on these ablation studies, we identified several key trade-offs. Increasing the number of candidates ($k$) is a highly effective way to improve accuracy with minimal impact on latency. The number of steps ($N$) shows diminishing returns after a certain point, and the masking schedule is sensitive, with gradual decay being optimal. Table 13 summarizes several high-performance configurations targeting different points on the speed-accuracy curve. For our main experiments reported in the paper, we selected the Accurate configuration ($k = 15$, $N = 4$) as it provides the best possible WER within a highly efficient latency profile.

Table 13: Selected high-performance configurations for PDD.

| Config | $k$ | $N$ | Schedule | WER (%) |
|---|---|---|---|---|
| Fast | 5 | 3 | [1.0, 0.9, 0.8] | 9.70 |
| Balanced | 10 | 4 | [1.0, 0.9, 0.85, 0.8] | 8.65 |
| Accurate | 15 | 4 | [1.0, 0.9, 0.85, 0.8] | 8.34 |

## G  QUALITATIVE EXAMPLES OF ITERATIVE REFINEMENT

We visualize the step-by-step evolution of individual tokens during inference. The following figures illustrate two key aspects of this process for several examples from the LibriSpeech test-clean set:

- **Token Finalization Process:** A grid showing at which step each token's prediction stabilizes and matches its final value for the remainder of the process.
- **Token Confidence Evolution:** A heatmap visualizing the model's confidence for each token at every refinement step.

These visualizations offer insights into how Whisfusion builds a transcript, rapidly committing to high-confidence tokens while iteratively refining more ambiguous parts of the sequence.

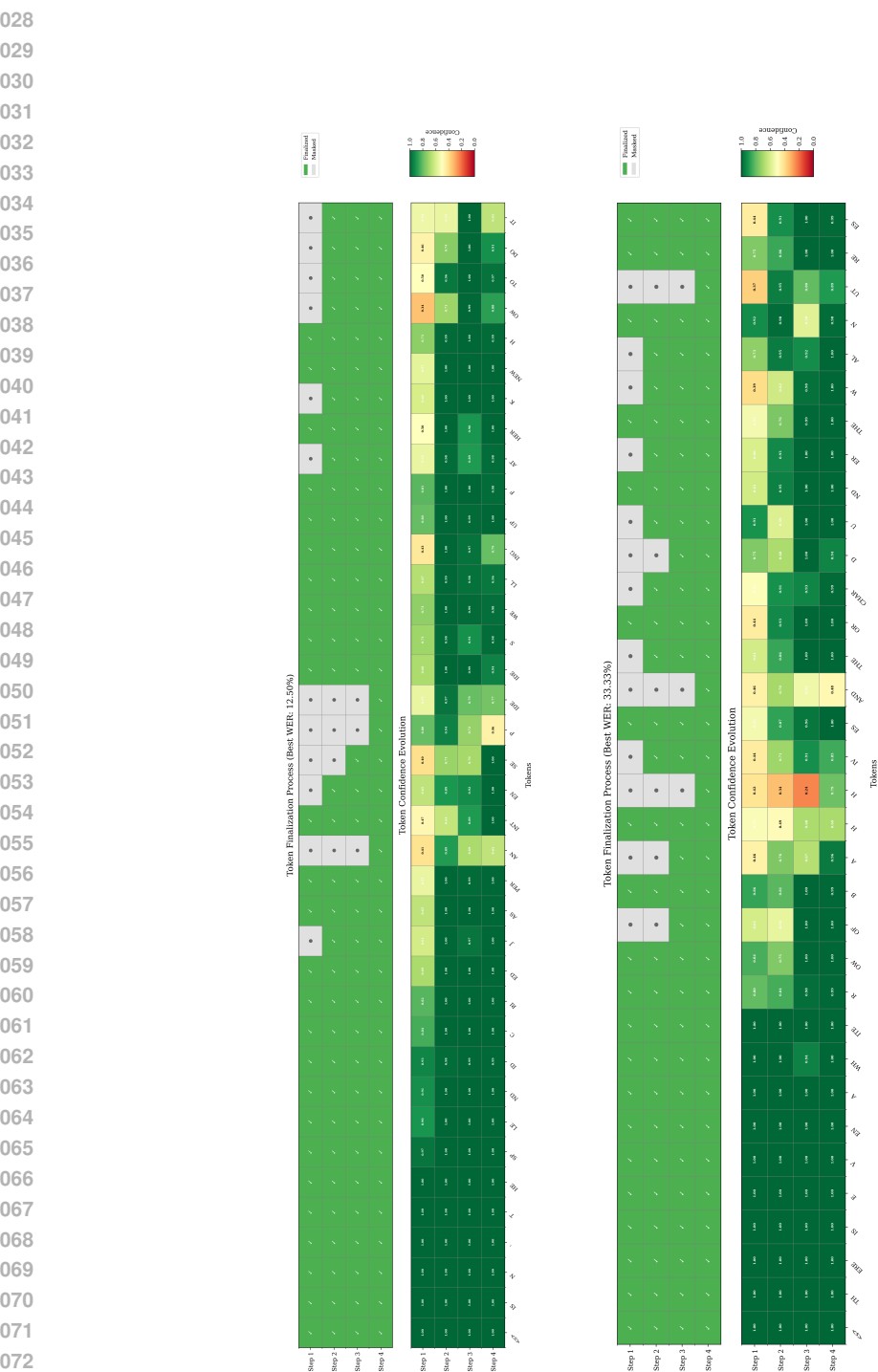

Figure 9: An additional example of the iterative refinement process. (rotated for readability)

