# OpenReview forum: "Whisfusion: Parallel ASR Decoding via a Diffusion Transformer"
_ICLR.cc/2026/Conference — Submitted to ICLR 2026_

### Official Review · Reviewer_Chhp · 2025-10-25

**Soundness:** 2
**Presentation:** 3
**Contribution:** 2
**Rating:** 2
**Confidence:** 4

**Summary:**

This paper presents Whisfusion, a non-autoregressive (NAR) automatic speech recognition (ASR) framework that fuses a pre-trained Whisper encoder with a masked diffusion text decoder via a lightweight cross-attention adapter trained using parameter-efficient fine-tuning (PEFT). The method improves the decoding speed of traditional autoregressive (AR) decoders by enabling parallel generation across the full acoustic context. A new Parallel Diffusion Decoding (PDD) strategy enables multi-candidate iterative refinement with negligible speed loss.
Fine-tuned solely on LibriSpeech (960h), Whisfusion achieves a lower WER than Whisper-tiny, and offers comparable latency on short audio. For longer utterances (>20s), it is up to 2.6× faster than the AR baseline.

**Strengths:**

- The paper is generally well-written.
- Improving the decoding speed of traditional AR decoders, by incorporating a masked diffusion text decoder is interesting.
- Using the cross-attention adapter trained via parameter-efficient fine-tuning (PEFT) to bridge the encoder and decoder is new.

**Weaknesses:**

- The paper does not correctly distinguish the decoding latency and the decoding speed. NAR decoding improves the decoding speed, but not the decoding latency. The encoder consumes the whole utterance, and then the NAR decoder outputs the entire text sequence and iteratively refine it (all positions in parallel).
- The latency of NAR decoding is a disadvantage, not suitable for streaming decoding. Streaming ASR performance is not evaluated for the proposed method.
- Whisfusion still lags behind Whisper-small (comparable in size) in WER, suggesting the approach has not yet closed the accuracy gap.
Compared to prior results trained solely on LibriSpeech (960h), the results of Whisfusion also lag behind.
https://github.com/thu-spmi/ASR-Benchmarks/blob/main/README.md#Librispeech
- Limited dataset scope. Results are confined to a LibriSpeech (clean English).
- The length estimation is a key challenge for NAR decoding. It is good to see the analysis of length estimation (but in appendix). The details of how Whisfusion performs length estimation are not clear.

**Questions:**

see above.

---

> ### Author Response · Authors · 2025-11-26
> **Response to Reviewer Chhp**
>
> We thank the reviewer for the thoughtful comments. We agree that clearer terminology and positioning are important, and we respond to each concern below with revisions and additional evidence.
>
> ## **W1 & W2. Distinguishing “latency” vs. “speed,” and suitability for streaming**
>
> **1. Clarification of terminology**
>
> We agree with your point that our use of the phrase “decoding latency” could be misunderstood as including time-to-first-token in a streaming/online setting. Whisfusion assumes an offline (batch) ASR scenario where the full utterance is first encoded and then decoded via multi-step NAR diffusion, and it is not designed to optimize streaming latency.
>
> In the revision, we will clarify the terminology more rigorously. Specifically, we will present our contributions as improvements in:
>
> - **utterance-level wall-clock decoding time / real-time factor (RTF)** in offline settings, and
> - **inference throughput** on the same hardware.
>
> Accordingly, we will avoid the ambiguous term “latency” as much as possible and replace it with explicit metrics such as “decoding time” and “RTF.”
>
> **2. Target scenario: non-streaming, turn-based ASR**
>
> Yes, Whisfusion is not intended for streaming, but it provides clear benefits in offline or turn-based interactive scenarios (e.g., voice assistants, smart speakers, robotics).
>
> - **AR bottleneck:** In AR models, user-perceived response time grows linearly with output length, creating increasing delay for longer utterances.
> - **Whisfusion advantage:** Whisfusion provides nearly constant wall-clock decoding time regardless of audio length (up to 30 s). This predictable and ultra-fast response after the user finishes speaking can significantly improve user experience in interactive systems.
>
> ---
>
> ## **W3. Performance gap to Whisper-Small**
>
> Your earlier concern that our initial LS-960h result (8.3% WER) lagged behind Whisper-Small (5.0%) was correct. As detailed in **General Response Section 1**, this gap was primarily due to data scarcity.
>
> - **Gap resolved:** After scaling the training data to **6.5k hours**, Whisfusion now achieves **4.9% WER** on LibriSpeech test-clean, effectively closing the accuracy gap while retaining its decoding-time advantage.
>
> ---
>
> ## **W4. Limited dataset scope**
>
> To address your concern about evaluation scope, we made the following changes:
>
> - **Training data expansion:** We increased the labeled training set to roughly **6.5k hours** (LibriSpeech + Libri-Light).
> - **Additional evaluation:** We evaluated on **TED-LIUM 3 (zero-shot)** and obtained 12.4% WER (see **General Response Table R3**).
>
> ---
>
> ## **W5. Details of length estimation**
>
> We apologize that the original description was not sufficiently clear.
>
> - **Mechanism:** Whisfusion does not use an explicit length predictor. Instead, it treats the **[EOS] token as a standard text token**, and during diffusion the model implicitly learns to place [EOS] at the correct position, thereby determining sequence length.
> - **Revision plan:** Our new experiments confirm that this mechanism works reliably even for longer utterances. In the final version, we will move the detailed explanation and analysis of this mechanism from the appendix into the main Methodology section.

---

### Official Review · Reviewer_J6px · 2025-10-27

**Soundness:** 4
**Presentation:** 3
**Contribution:** 4
**Rating:** 10
**Confidence:** 5

**Summary:**

Authors present strategy for finetuning whisper models with diffusion for ASR tasks. This is achieved with two stage training strategy with pretrained models. In addition, they present a parallel decoding strategy that takes advantage of diffusion based inference for additional gains.

**Strengths:**

Architecture is suitably novel that it contributes to ASR research, and the use of diffusion models in lieu of autoregression is an active field of discussion for general ML research. Given the acceptable performance on a common benchmark, these two aspects alone merit acceptance.

**Weaknesses:**

Proposed model uses pretrained checkpoints for SFT while autoregressive models can be trained from scratch. It would be useful if authors can present information on full pipeline training with current model instead of use of pretrained checkpoints.

Training was performed on a relatively easy benchmark (LS is read speech from audio-book actors). This makes it unknown how well diffusion approaches work for real-world speech recognition cases (telephony, conversational speech, high noise presence).

**Questions:**

Requests from authors for camera ready submission:
- If possible, please provide additional evaluation numbers on non-librispeech benchmarks. No need to train whole model, but having deltas from CommonVoice, Fisher, and the like would help show how model responds to domain drift. These can be provided in appendix.
- Move discussion of diffusion loss from appendix to main body of work. Diffusion is less common in speech recognition and readers would benefit from review.
- Resize image on page 20, it's not even close to readable as currently presented.
- Was there an attempt to train from scratch? Even if results are paltry, there would be community benefit to acknowledge if there are limitations on diffusion in full pipeline training.

---

> ### Author Response · Authors · 2025-11-26
> **Response to Reviewer J6px**
>
> Thank you for your supportive review and for recognizing the novelty and potential impact of our work. Your suggestions for the camera-ready version are very helpful, and we believe they will improve the paper substantially. We respond to each point below.
>
> **1. Additional benchmarks beyond LibriSpeech**
>
> We fully agree that evaluating domain drift is important.
>
> - **Current progress:** As a first step, we conducted evaluation on **TED-LIUM 3 (zero-shot)**. As shown in **General Response Table R3**, Whisfusion achieves 12.4% WER, which is comparable to the CTC baseline (11.7%) and close to the AR Whisper decoder (10.1%), demonstrating robust generalization.
> - **Commitment:** Following your suggestion, we plan to add results on additional benchmarks and include them in the appendix of the camera-ready version to further validate robustness.
>
> **2. Moving the diffusion loss explanation to the main text**
>
> This is an excellent suggestion. For better readability and accessibility to the ASR community, we will move the explanation and formulation of the diffusion loss from the appendix into the Methodology section.
>
> **3. Adjusting figure size**
>
> We apologize for the reduced readability of the figures on page 20. In the final manuscript, we will enlarge these figures and improve their resolution for clearer presentation.
>
> **4. Attempting from-scratch training**
>
> We agree with your view that verifying the limits of training from scratch is valuable to the community. We are currently preparing this experiment and will make our best effort to include the results in the camera-ready version regardless of whether they are positive or negative. In addition, we plan to open-source all code and training scripts so that the community can further build upon and extend this work.

---

### Official Review · Reviewer_RnDy · 2025-10-31

**Soundness:** 2
**Presentation:** 3
**Contribution:** 2
**Rating:** 4
**Confidence:** 3

**Summary:**

This paper proposes Whisfusion, a novel non-autoregressive (NAR) ASR framework that combines a pre-trained Whisper encoder with a diffusion-based text decoder. The two components are connected via a lightweight cross-attention adapter trained through parameter-efficient fine-tuning . The authors also design a batch-parallel multi-step decoding strategy that improves recognition accuracy while maintaining efficiency. Experiments on LibriSpeech (960h) show that Whisfusion outperforms Whisper-tiny and achieves up to 2.6× faster decoding speed for long-form utterances.

**Strengths:**

Novel architecture: Using a diffusion transformer as a decoder for ASR is highly original and demonstrates an interesting fusion of generative modeling and speech recognition.

Effective integration: The cross-attention adapter provides a clear and efficient way to connect the Whisper encoder and the diffusion decoder.

Improved decoding strategy: The proposed multi-step batch decoding (similar to beam search) is a well-motivated idea that improves the quality-speed trade-off compared to previous NAR models that rely solely on argmax decoding.

Clear experimental gains: On LibriSpeech, Whisfusion achieves measurable WER improvements over Whisper-tiny, showing the potential of diffusion-based decoding in ASR.

**Weaknesses:**

Limited performance improvement: Although Whisfusion uses the Whisper-small encoder, its performance still lags significantly behind the Whisper-small model, raising questions about the effectiveness of the diffusion-based decoding in capturing linguistic dependencies.

Lack of text length modeling: The model does not explicitly predict or control text length, which could lead to substantial computational waste, especially for short utterances.

Limited comparisons: The paper does not compare with other recent non-autoregressive ASR baselines , making it difficult to contextualize the contribution.

Single dataset evaluation: All experiments are conducted only on LibriSpeech; results on other datasets (e.g., TED-LIUM, CommonVoice) would make the conclusions more convincing.

**Questions:**

Have you considered using adaptive text length prediction to reduce unnecessary computation?

Could Whisfusion benefit from fine-tuning both encoder and adapter jointly on downstream datasets?

---

> ### Author Response · Authors · 2025-11-26
> **Response to Reviewer RnDy**
>
> We sincerely thank the reviewer for the insightful feedback. Following your suggestions, we conducted substantial additional experiments and revised the manuscript to clarify our contributions and positioning.
>
> ## **W1 & W3 & W4 (Performance, Baselines, Datasets)**
>
> We sincerely thank you for your insightful feedback. Through the large-scale additional experiments detailed in the General Response, we have addressed the key concerns regarding performance limitations (W1), lack of strong baselines (W3), and evaluation on a single dataset (W4).
>
> - **Performance & Data Scaling:** By expanding the training data to **6.5k hours**, Whisfusion now reaches **Whisper-Small-level accuracy** (LibriSpeech test-clean WER **4.9% vs. 5.0%**), and the long-form degradation observed in the initial submission is resolved.
> - **Baselines (CTC Comparison):** We implemented a strong CTC baseline under identical conditions (sharing the Whisper encoder). While CTC achieves slightly lower WER numerically (4.3% vs. 4.9%), we argue that Whisfusion’s central contribution lies in its **architectural nature as a generative framework** (see General Response Sec. 4). Unlike CTC, which is restricted to **discriminative alignment**, Whisfusion preserves the **generative capability of AR Transformers**, enabling natural extensions to more complex tasks such as speech translation and instruction-following.
> - **Datasets:** We additionally evaluated on **TED-LIUM 3**, confirming that Whisfusion exhibits robust zero-shot generalization beyond LibriSpeech.
>
> ---
>
> ## **W2 & Q1: Adaptive Text Length Prediction**
>
> Your concern about potential computation waste due to the absence of explicit length prediction is well taken. We acknowledge that processing padded sequences can introduce redundant computation.
>
> - **A Trade-off for Stability:** In early experiments, we explored adaptive-length strategies (e.g., predicting length first and adjusting the masking window accordingly). However, we observed that such approaches introduced a mismatch between training (fixed-length) and inference (variable-length), leading to **instability and performance degradation**.
> - **Design Choice:** Consequently, we prioritized stability and simplicity. By letting the model implicitly predict the [EOS] position within a fixed buffer—without an auxiliary length predictor—we achieved more reliable convergence. Given GPU batch parallelism, we believe this trade-off is favorable for ensuring a robust generation process.
>
> ---
>
> ## **Q2: Joint Fine-tuning of Encoder and Adapter**
>
> We did conduct preliminary experiments on joint fine-tuning, but ultimately chose not to adopt it for the following reasons:
>
> - **Training Instability:** When unfreezing the encoder, training became highly sensitive to hyperparameters, especially the encoder learning rate, frequently resulting in instability or dataset-specific overfitting.
> - **Research Goal (Decoder Optimization):** Our primary objective is to optimize a non-autoregressive (NAR) decoder that works effectively with a powerful, well-established encoder (Whisper). Freezing the encoder aligns with the PEFT (parameter-efficient fine-tuning) philosophy, ensuring that the decoder learns to align to robust, general-purpose acoustic features rather than relying on encoder adaptation to a specific downstream task.

---

### Official Review · Reviewer_haAb · 2025-10-31

**Soundness:** 1
**Presentation:** 2
**Contribution:** 1
**Rating:** 2
**Confidence:** 5

**Summary:**

A text diffusion decoder is extended by cross-attention to an audio encoder, to build a non-autoregressive speech recognition model using diffusion.

The motivation is to improve the recognition speed.

The authors use the pretrained Whisper encoder and leave it frozen through all the experiments. The authors use a pretrained  mdm-170M diffusion LM.

There are two training stages: In the first stage, only the added cross-attention part is trained. In the second stage, the whole decoder is trained.

Librispeech is used for training (finetuning) and also for evaluation. As the authors use pretrained models, a lot more data was implicitly used.

Their final Whisfusion model performs better than Whisper-tiny, but worse than all other Whisper variants. All the Whisper variants were taken as-is and not fine-tuned on Librispeech.

**Strengths:**

* Using diffusion for ASR is an interesting and promising direction.
* Good speedups for recognition.
* Study on the parallel diffusion decoding (PDD) is interesting and shows its importance.

**Weaknesses:**

* The experimental setting is bad. We cannot really learn much about the most relevant questions (how does such a diffusion model compare to other alternative ASR models, under the same conditions, same training data that was used implicitly or explicitly).
* Phrasing it as novel is not totally correct.
* It is sold as a good solution specifically for long-form ASR, but then the experiments show that this is very it performs really bad.
* No real scaling laws analysis.
* Analysis should be extended.

See comments/questions below for more details on these points.

**Questions:**

Abstract " We propose Whisfusion, the first framework to fuse a pre-trained Whisper encoder with a text diffusion decoder." - I'm not sure what "first" refers to. This very specific thing, that a pre-trained Whisper encoder was used specifically? Or the more generic aspect that pretrained models were used to get a diffusion ASR model? Or does this say that this is the first diffusion ASR model? The latter is definitely not true. For example, there is:
- TransFusion: Transcribing Speech with Multinomial Diffusion, https://arxiv.org/abs/2210.07677
- Cross-Modality Diffusion Modeling and Sampling for Speech Recognition, https://www.isca-archive.org/interspeech_2024/yeh24_interspeech.pdf
- Audio-Conditioned Diffusion LLMs for ASR and Deliberation Processing, https://arxiv.org/abs/2509.16622 (to be fair, very recent...)
- Drax: Speech Recognition with Discrete Flow Matching, https://arxiv.org/abs/2510.04162 (ok, to be fair, very recent. but check its related work)

Abstract "Fine-tuned solely on LibriSpeech (960h), Whisfusion achieves a lower WER than Whisper-tiny (8.3% vs. 9.7%)" - these numbers are not at all impressive... There are much simpler pure CTC models (i.e. also non-autoregressive, but even much faster than diffusion) with much better numbers.

"Novel NAR Framework" - I think describing it as novel is overselling it. A couple of similar models have been proposed before. See above.


No good comparison on top of Whisper: E.g finetune Whisper on Librispeech. Or finetune a CTC on top of the Whisper encoder on Librispeech. Or finetune another smaller/simpler decoder on top of the Whispr encoder on Librispeech. Or finetune some LLM on top of the Whisper encoder.


"batch-parallel, multi-step decoding ... minimal impact on speed" - I assume this is only the case because you parallelize it, assuming you have enough cores idling around? Let's say you already fully utilize the GPU before (e.g. because of batched decoding multiple audio files in parallel), then this would have a big impact on speed?


"... establishing a new, efficient operating point for long-form ASR"  - but is it actually performing bad for longer audio? (Table 3)

Table 3: Usually you mark the best results in bold. Here you just mark Whisfusion results always in bold, which is thus misleading.

Multiple different model sizes should be tested, for some scaling law analysis.

Model performs bad on long-form. Why? Any analysis? Maybe could be better with better decoding? But then again would probably be slower.


What do we really learn from this? Just that it works at all? But that was to be expected, and that was also already shown before. What else? The WER comparison to other models would be most interesting, but in its current form (Table 3, Table 4), it is somewhat meaningless, as the training data in each case is different.

Sec 5.2.1 "”w/o Acoustic Conditioning1“" - what is "Conditioning1"? Typo?

Sec 5.2.1 / Table 5: I don't quite understand the w/o acoustic conditioning experiment: In what stage do you remove this? Stage 1? Stage 2? Both stages? Do you even do both stages here, or only stage 1? I assume you remove it for both stages, and then also don't use it in recognition? I assume you anyway did both training stages? So basically you finetuned the diffusion LM just as a LM without conditioning? "Despite masking only 30% of the tokens from the ground truth transcript" - what do you mean by that? You mask 30%, and wanted to see how good the model can recover it, just as a LM, without conditioning? Again, to be clear: Here, you also trained the model without conditioning? So then the result (150.8% WER) is unexpected, right? You would maybe expect 30% or less. Either I don't fully understand the experiment, or sth is broken here. Did you analyze this? What does the model generate as output?


"we .. propose Parallel Diffusion Decoding (PDD), a novel inference strategy ..." - It's not really so novel. Similar ideas have been done for diffusion models before. For example (mostly citing only the recent ones, but there are many older ones as well):
- https://arxiv.org/abs/2306.17775
- https://arxiv.org/abs/2503.02039
- https://arxiv.org/abs/2505.24857
- https://arxiv.org/abs/2508.08712 (it's a survey, it cites many earlier works)
- https://arxiv.org/abs/2509.25188
- https://arxiv.org/abs/2509.23146
- https://arxiv.org/abs/2510.21961 (ok, to be fair, that's very recent, but see their related work)


There also should be experiments for not importing pretrained models, for comparison.

**Details Of Ethics Concerns:**

.

---

> ### Author Response · Authors · 2025-11-26
> **Response to Reviewer haAb**
>
> We sincerely thank you for your detailed and critical review.
>
> As described in our General Response, we strengthened the experimental evidence by **scaling the training data to 6.5k hours** and by **re-implementing and re-training both CTC and AR baselines from scratch under identical conditions**.
>
> In the responses below:
>
> - **W#** follows the numbering of the *Weakness* items you listed, and
> - **Q#** follows our own topic-grouped ordering of your individual questions.
>
> Each Q item is titled with a short summary of its main point, and we answer related questions jointly to avoid redundancy while keeping the responses clear and complete.
>
> ## **Section 1. Experimental Setup & Baselines (Response to W1, Q3, Q10)**
> ### **Fair Comparison with Baselines (W1, Q3)**
> Your point that we should compare against strong NAR baselines such as CTC under identical conditions is entirely valid, and we have directly reflected this in the revision. Specifically, under the same setup as Whisfusion (sharing the Whisper encoder and trained on the same 6.5k h dataset), we newly implemented and trained:
>
> - an **AR Whisper-style decoder (from scratch)**, and
> - a **CTC-based NAR decoder (from scratch)**.
>
> The resulting performance on LibriSpeech test-clean is as follows (see General Response Table R3 for full settings and results):
>
> - Whisper-Small (public model, 680k h): 5.0% WER
> - AR Whisper Decoder (6.5k h, scratch): 5.3% WER
> - CTC Baseline (6.5k h, NAR): 4.3% WER
> - Whisfusion (6.5k h, NAR): 4.9% WER
>
> Thus, as you anticipated, the CTC baseline achieves a lower WER numerically as an ASR-specialized model.
>
> However, the goal of this work is not to propose yet another minimal NAR-ASR head, but rather to explore a **Generative NAR decoder that can replace the AR decoder while preserving Whisper’s generative Transformer structure**. A detailed discussion on output quality, contextual integrity, and extensibility at comparable WER ranges is provided in **General Response Sections 3–4**.
>
> ### **Training from Scratch (Q10)**
>
> We agree that evaluating models without pretraining is important for understanding the intrinsic potential of an architecture.
>
> That said, our central objective is to test the fusion/transfer setting itself—namely, *how effectively a strong pre-trained audio encoder (Whisper) can be fused with a pre-trained masked-diffusion text LM*. Therefore:
>
> - We shared the same pre-trained Whisper encoder across all comparison models, so that the effects of decoder design and fusion strategy could be isolated in a fair manner; and
> - Given that Whisper is trained on hundreds of thousands of hours, it is practically difficult in an academic setting to build competitive scratch-trained models at comparable scale.
>
> Looking forward, once larger data and compute become available, we see it as an important extension to train Whisfusion fully from scratch and verify whether its scaling advantages hold even without transfer.

---

> > ### Comment · Reviewer_haAb · 2025-11-27
> >
> > > we strengthened the experimental evidence by scaling the training data to 6.5k hours and by re-implementing and re-training both CTC and AR baselines from scratch under identical conditions.
> >
> > What do you mean by "identical conditions"? My point was that you want to compare the proposed diffusion model to some other ASR baselines. So why do you retrain CTC?
> >
> > By AR baseline, you mean the Transformer AED model for ASR?
> >
> > I want to have a table somewhere, like:
> >
> > | Model | WER |
> > | ---- | ---- |
> > | CTC | ... |
> > | AED | ... |
> > | Diffusion | ... |
> >
> > And those three models are trained exactly in the same way, i.e. all from scratch, using the same amount of training data, not importing any pretrained models.
> >
> > If your point is that you can import an existing model, anyway do such from-scratch experiments, and then additional do such an import experiment, but also do a similar comparison experiment for CTC or AED, such that you have a proper comparison.

---

> > > ### Author Response · Authors · 2025-12-04
> > > **Follow-up Comments: Clarifying the Core Motivation and Problem Setting**
> > >
> > > Thank you again for your active and detailed feedback.
> > >
> > > Your five follow-up comments are closely related, so we would like to respond to them in an integrated way, focusing on **the core motivation of our work and the intended problem setting.**
> > >
> > > ## **1. Experimental Design and Why We Retrained CTC**
> > >
> > > When we wrote “identical conditions,” we meant the following:
> > > - **Same speech encoder, same training data, same training setup; only the decoder is changed:**
> > > 	- AR Transformer decoder (Whisper-style)
> > > 	- CTC decoder
> > > 	- Our diffusion decoder (Whisfusion)
> > >
> > > We added this comparison because, in your original review, you suggested:
> > >
> > > > “... finetune a CTC on top of the Whisper encoder on LibriSpeech, ...”
> > >
> > > which we interpreted as a request that **Whisper-top baselines (CTC/other decoders on top of a frozen Whisper encoder) are needed.**
> > >
> > > We therefore added experiments aligned to that interpretation.
> > >
> > > In your follow-up comments, we now understand that you would like to see:
> > >
> > > > “CTC / AED / Diffusion all trained from scratch”
> > >
> > > on the same data, without using any pretrained models.
> > >
> > > This targets a different question from the one above, which assumes a transfer/fusion setting with a pretrained Whisper encoder.
> > >
> > > The primary research question of our paper is:
> > >
> > > > “Given an already well-trained Whisper-style encoder,
> > > **can we replace the AR decoder while preserving architectural compatibility with Whisper,**
> > > and **achieve similar ASR accuracy with significantly reduced decoding time?**”
> > >
> > > For this reason, in the current revision we focused on ensuring a fair comparison within this transfer/fusion setting.
> > >
> > > We fully agree that an additional study where CTC, AED, and diffusion models are all trained from scratch on the same dataset, without any pretrained components, would provide a complementary, architecture-centric perspective.
> > >
> > > However, it lies on a different axis from our main focus:
> > >
> > > - **Our current focus:** “Whisper-based fusion/acceleration” with a shared pretrained encoder
> > > - **Your new suggestion:** architecture-only comparison without any pretrained components
> > >
> > > Due to resource and space limitations, we were not able to fully cover both axes in a single submission.
> > > We see this scratch-only comparison as a natural follow-up study and would like to pursue it in future work.
> > >
> > > ## **2. What We Mean by “Preserving Whisper’s Generative Transformer Structure”**
> > >
> > > In your follow-up comments, you raised two closely related questions:
> > >
> > > > “And also, why is the CTC baseline better? Wouldn't you expect that the diffusion model should still be better than CTC?”
> > >
> > > and
> > >
> > > > “Also, you write that CTC cannot do these tasks (Speech Translation, Instruction Following / Multi-tasking, Simultaneous Multilingual Output). I'm not sure this is actually really the case. I think CTC might be able to do that.”
> > >
> > > We would like to clarify what we mean by *“preserving Whisper’s generative Transformer structure”* and why we view this as important beyond **pure-ASR WER**.
> > >
> > > We would like to clarify what we intended to convey and the technical background behind it.
> > >
> > > - **Structural limitations of CTC (Structural Limitation)**
> > >
> > >   CTC is mathematically based on a **monotonic alignment assumption**.
> > >
> > >   That is, it assumes that the temporal order of the input speech and the output text should match.
> > >   In other words, the model relies on monotonic speech–text alignment.
> > >
> > >   Because of this, for tasks such as
> > >   - translation, where word order may change, or
> > >   - instruction following / summarization, where the model may need to generate content not directly aligned to specific input frames,
> > >
> > >   a pure CTC-based architecture often requires **substantial architectural modifications**, or must accept a **nontrivial performance trade-off**.
> > >
> > > - **Strategic value of Whisfusion**
> > >
> > >   In contrast, Whisfusion fully inherits **Whisper’s generative Transformer structure** (Self-Attention / Cross-Attention).
> > >
> > >   **Only the sampling scheme is changed from AR to masked diffusion; the overall decoder form is preserved.**
> > >
> > >   This means that, without any architectural change, Whisfusion can enable translation, summarization, and multi-task generation in a native way, just as in Whisper.

---

> > > > ### Author Response · Authors · 2025-12-04
> > > > **Follow-up Comments: Clarifying the Core Motivation and Problem Setting (Continued)**
> > > >
> > > > ## **3. On the Term “Long-form ASR”, the 20–30s Regime, and Data Requirements**
> > > >
> > > > As you rightly pointed out,
> > > > “long-form ASR” often refers to utterances of several minutes to hours.
> > > >
> > > > From that perspective, using the term “long-form ASR” in our initial abstract was indeed too strong, and we agree with this criticism.
> > > >
> > > > In our actual setting:
> > > > - Our model is limited by the Whisper encoder architecture to a maximum of ~30 seconds per input chunk.
> > > > - Within this 0–30s window, we used the 20–30s bucket as a relatively “long” utterance regime for analysis.
> > > >
> > > > Thus, the “long utterances” in the current paper refer specifically to long segments within Whisper’s 30s window, not multi-minute or hour-long recordings. We will reflect this more precisely in the final wording.
> > > >
> > > > You also raised the concern that:
> > > >
> > > > > "... it looks like a disadvantage of Whisfusion needs more training data to be able to handle that ..."
> > > >
> > > > Our interpretation and response are as follows:
> > > > - We do not believe that Whisfusion is **inherently more data-hungry** than other architectures.
> > > > - The degradation in the 960h setting is, in our view, primarily due to the **extreme sparsity of long utterances** in LibriSpeech 960h:
> > > >   - utterances longer than 20 seconds account for **less than 0.01%** of the training data.
> > > >   - This makes it very hard for a NAR decoder to reliably learn length/EOS behavior for 20–30s segments.
> > > > - In the 6.5k h setting, where we explicitly add data with more 20–30s utterances, we observe:
> > > >   - a **strong recovery** in the 20–30s bucket, and
> > > >   - a **steep improvement** in overall performance.
> > > >
> > > > The estimated scaling exponent ($k \approx 0.245$) suggests that the model is still in a **non-saturated, data-efficient regime**, and that the architecture has room to further converge toward AR performance as data grows.
> > > >
> > > > More broadly, Transformer-based models are known to follow scaling laws where **large amounts of data are required**, but performance continues to improve steadily as data increases.
> > > > Whisfusion shares this property: it is designed to leverage large-scale data in a similar way, while providing the additional benefit of **NAR decoding speed** once the accuracy has been brought close to the AR baseline.
> > > >
> > > > In particular, under the 6.5k h setting, Whisfusion maintains accuracy comparable to Whisper-small while reducing decoding time on 20–30s segments from **674.7 ms to 80.7 ms (8.4× faster)**.
> > > >
> > > > Our goal in this work is to show that, under such a regime, it is possible to approach Whisper-small–level accuracy while **significantly reducing decoding time**, and at the same time retain the generative, multi-task–friendly structure of the Whisper decoder.

---

> > ### Comment · Reviewer_haAb · 2025-11-27
> >
> > > Thus, as you anticipated, the CTC baseline achieves a lower WER numerically as an ASR-specialized model.
> > >
> > > However, the goal of this work is not to propose yet another minimal NAR-ASR head, but rather to explore a Generative NAR decoder that can replace the AR decoder while preserving Whisper’s generative Transformer structure.
> >
> > I don't really understand this.
> >
> > "while preserving Whisper’s generative Transformer structure" - what do you mean by that? And why is that a nice thing to have? What's the point of that?
> >
> > If there is some other point despite ASR, it would be good to actually demonstrate that. E.g. maybe you are thinking about speech translation, or some dialogue model, or some other application?
> >
> > And also, why is the CTC baseline better? Wouldn't you expect that the diffusion model should still be better than CTC?

---

> > ### Comment · Reviewer_haAb · 2025-11-27
> >
> > > That said, our central objective is to test the fusion/transfer setting itself—namely, how effectively a strong pre-trained audio encoder (Whisper) can be fused with a pre-trained masked-diffusion text LM. Therefore:
> >
> > > Given that Whisper is trained on hundreds of thousands of hours, it is practically difficult in an academic setting to build competitive scratch-trained models at comparable scale.
> >
> > I understand that. But you don't need to do this at comparable scale. Chose some reasonable scales that you can handle. For example, just train on Librispeech.
> >
> > This is just one other baseline. But there you can fully control all the conditions. That allows for much more direct comparisons.
> >
> > You can also keep the Whisper encoder for other experiments. But then there should be some reasonable comparison to what you do. E.g. then train CTC on top of the Whisper encoder. And train the diffusion decoder from scratch, to have that comparable. And/or also train a normal LM from scratch in the same way, to have some proper comparison.

---

> ### Author Response · Authors · 2025-11-26
> **Response to Reviewer haAb (Continued)**
>
> ## **Section 2. Performance & Long-form (W3, Q2)**
> ### **On long-form performance (W3)**
> We fully agree with your concern that, in the initial submission trained only on LS-960h, Whisfusion appeared unsuitable for long-form ASR due to the severe degradation on 20–30s segments (WER rising to 15.9%). However, we believe this issue was not a fundamental architectural limitation, but rather a consequence of **extreme long-utterance sparsity and skewed data distribution** in LS-960h (utterances longer than 20s account for **<0.01%** of the training data).
>
> To verify this hypothesis, we scaled the training data to **6.5k hours** and retrained the same architecture. After scaling, the WER for the 20–30s range improved dramatically from **15.9% → 4.8%**, and the duration-wise performance became stable across all length bins. This confirms that the long-form collapse was **data-scale/distribution–driven**, and that with sufficient long-utterance coverage, Whisfusion can maintain consistent performance even on long-form inputs. (See **General Response Table R2 / Section 1**)
>
> ### **On comparisons with CTC and Whisper-tiny (Q2)**
> You correctly observed that simpler NAR models, such as CTC, can achieve numerically superior results in pure ASR tasks. Our new controlled experiments confirm this: under identical conditions (shared Whisper encoder, 6.5k-hour training), the CTC baseline achieves a marginally lower WER (4.3%) compared to Whisfusion (4.9%) on LS test-clean (see **General Response Table R3**).
>
> However, we argue that the research value of Whisfusion extends beyond a single WER metric, lying instead in its architectural potential:
>
> 1. **Competitive accuracy at scale:** With 6.5k hours, Whisfusion reaches **Whisper-Small–level WER** (4.9% vs. 5.0%), and substantially improves over Whisper-Tiny (9.7%).
> 2. **Extensibility of a generative decoder:** Unlike CTC, Whisfusion preserves the **Whisper-style generative Transformer decoder structure** in a fully NAR framework. Our goal is to retain long-context generative modeling and enable natural extensions to generative speech–text tasks such as speech translation or instruction following (discussed in **General Response Section 4**).
> 3. **Qualitative differences in output:** As shown in **General Response Section 3**, even at similar WER ranges, Whisfusion tends to better preserve sentence-level form and context, producing more recoverable, structurally sound outputs than CTC.
>
> In short, while CTC can yield slightly lower WER as an ASR-specialized head, Whisfusion targets a different operating point: a **“Generative NAR Decoder”** that achieves NAR speed **while inheriting Whisper’s generative architectural strengths**. A more complete quantitative/qualitative comparison and positioning is provided in **General Response Sections 2–4**.

---

> > ### Comment · Reviewer_haAb · 2025-11-27
> > **Re long-form ASR**
> >
> > Note on long-form ASR: I think this is still not really long-form. In common long-form ASR settings, you would evaluate it for several minutes, sometimes even hours.
> >
> > But it's good to see that it recovers with more training.
> >
> > But then on the other side, it looks like a disadvantage of Whisfusion needs more training data to be able to handle that, while your other baselines are fine with less training data? (Or do you actually have the proper comparison here?)

---

> > ### Comment · Reviewer_haAb · 2025-11-27
> >
> > > Whisfusion tends to better preserve sentence-level form and context, producing more recoverable, structurally sound outputs than CTC.
> >
> > How do you measure/quantify that?
> >
> > > Beyond What CTC Can Reach
> >
> > I see the point in that. But:
> >
> > * Maybe you should show this? (Although, maybe that would go a bit beyond the scope. Not sure...)
> > * Still, why is CTC actually better for ASR? I don't see a good reason for this.
> >
> > Also, you write that CTC cannot do these tasks (Speech Translation, Instruction Following / Multi-tasking, Simultaneous Multilingual Output). I'm not sure this is actually really the case. I think CTC might be able to do that. So, I'm not really sure whether a diffusion model would really perform better on these tasks compared to a CTC model.

---

> ### Author Response · Authors · 2025-11-26
> **Response to Reviewer haAb (Continued)**
>
> ## **Section 3. Novelty & Positioning (W2, Q1, Q4, Q9)**
> ### **W2 & Q1. Novelty / the “first” claim and positioning within prior work**
> First, we acknowledge your point that the sentence in our abstract,
>
> > “… the first framework to fuse a pre-trained Whisper encoder with a text diffusion decoder.”
>
> in its current form can be read as claiming the first diffusion-based ASR, which would be an over-claim given prior diffusion ASR works such as TransFusion. We clearly agree with this concern.
>
> What we originally intended by “first” was narrower. To the best of our knowledge, Whisfusion is the first framework that
>
> - takes a **pre-trained masked-diffusion text LM** (MDM family),
> - fuses it with a Whisper encoder via a **lightweight cross-attention adapter**, and
> - forms a **fully non-autoregressive (fully NAR) ASR decoder**.
>
> However, since this nuance is easy to miss and may cause misunderstanding, we will remove the word “first” from the abstract and replace it with a neutral statement, e.g.,
>
> > “We propose Whisfusion, a framework that fuses a pre-trained Whisper encoder with a text diffusion decoder via a lightweight cross-attention adapter.”
>
> We will also revise the related-work section to more systematically clarify the relationship between prior diffusion ASR and Whisfusion.
>
> ---
>
> ### **Differences from TransFusion / Cross-Modality Diffusion ASR**
>
> We agree that the works you mentioned are important foundations for diffusion-based ASR. That said, Whisfusion follows a distinct line of approach, differing both in its diffusion paradigm and in how it integrates pre-trained models.
>
> - **Diffusion type**
>     - TransFusion / Cross-Modality: **multinomial (D3PM / DDPM-style) discrete diffusion**
>     - Whisfusion: **masked diffusion (MDM)** with **multi-step [MASK]-infilling**
>
>     Intuitively, the former rewrites the entire sentence at every step while gradually reducing randomness, whereas Whisfusion is closer to a **masked-LM-style diffusion that progressively fills only the blanks**.
>
> - **Acoustic conditioning mechanism**
>     - Prior models inject audio into an ASR-from-scratch diffusion decoder, typically by **implicit self-attention coupling** (concat/add of audio tokens or vectors) or variant cross-attention designs.
>     - Whisfusion **preserves the Whisper encoder–decoder form**, and inserts **lightweight cross-attention adapters** into each block of a pre-trained text MDM to align acoustic information.
>
>     A key distinction is that Whisfusion targets a **NAR generative decoder structurally aligned with the Whisper decoder**.
>
> - **Use of pre-trained diffusion LMs**
>     - Prior models: the diffusion decoder is effectively trained from scratch for ASR.
>     - Whisfusion: directly reuses a pre-trained masked-diffusion text LM (mdm-170M) and efficiently bridges it to Whisper via a lightweight adapter and a two-stage curriculum.
>
>     Rather than proposing yet another ASR-specific diffusion decoder, our focus is on **how to efficiently fuse two large pre-trained models (audio + text)**.
>
>
> Because these distinctions were not sufficiently emphasized in the original draft, we will strengthen the related-work and model-description sections accordingly.
>
> ---
>
> ### **Relation to Audio-Conditioned Diffusion LLMs / Drax**
>
> We also view "Audio-Conditioned Diffusion LLMs for ASR and Deliberation Processing" and "Drax: Speech Recognition with Discrete Flow Matching" as closely related contemporary/follow-up works that appeared after Whisfusion.
>
> - **Audio-Conditioned Diffusion LLMs**
>     - Combines a Whisper-Large–style encoder with a 8B diffusion LLM, focusing on large-scale deliberation / multi-step diffusion.
>     - Acoustic conditioning is extracted via a Q-Former + projection and injected by concatenating with text tokens.
>     - In contrast, Whisfusion uses a ~300M **ASR-specialized masked-diffusion LM**, attaches an **explicit cross-attention adapter**, and targets 4-step fully NAR decoding—so both the scale and the goal differ substantially.
> - **Drax**
>     - Uses a Whisper-Large–style encoder and a DiT-style decoder under a **discrete flow matching** framework.
>     - Its generation process is based on **flow matching rather than diffusion**, and its audio-conditioning pathway is also notably different from Whisfusion.
>
> Since both papers were released after our original submission, we could not cite them then. In the camera-ready version, we will include both works and clarify Whisfusion’s position in the broader diffusion/flow ASR landscape, highlighting that it is characterized by
>
> - a **masked-diffusion text decoder**, and
> - a **lightweight cross-attention fusion strategy**,
>
> and thus occupies a distinct point in the family of diffusion/flow-based ASR approaches.

---

> > ### Author Response · Authors · 2025-11-26
> > **Continuing Section 3**
> >
> > ### **Q4 & Q9. On the Speed Impact and “Novelty” of Parallel Diffusion Decoding (PDD)**
> > As you rightly pointed out, the phrases in our draft—“minimal impact on speed” and "novel inference strategy”—were overly strong and did not sufficiently reflect the broader diffusion literature. In the revision, we will make both claims more accurate and explicitly conditional.
> >
> > **(1) On the speed claim (“Minimal Impact on Speed”)**
> >
> > We agree with your observation that in throughput-oriented, large-batch regimes where GPUs are already saturated, increasing the number of diffusion steps k inevitably reduces throughput.
> >
> > Our intended point was narrower: in low-to mid-batch settings, such as real-time ASR, where idle GPU parallelism remains, PDD can leverage batch-parallel decoding to improve the WER–decoding-time (RTF) trade-off with only modest additional wall-clock overhead.
> >
> > To avoid ambiguity, we will revise the statement to explicitly include this condition, e.g.,
> >
> > > “In low- to mid-batch regimes with available GPU parallelism, PDD yields only modest wall-clock overhead while providing a better  WER–decoding-time (RTF) trade-off.”
> >
> > **(2) On the novelty of PDD**
> >
> > We acknowledge that the papers you cited (e.g., Wu et al., 2023, and subsequent work on conditional/parallel sampling and guidance) are highly relevant prior studies. Many of them run multiple trajectories in parallel or incorporate SMC/importance structures, improving the sample-quality vs. compute trade-off through parallelism—which clearly overlaps with the high-level idea behind PDD.
> >
> > Given this context, we agree that calling PDD a “novel inference strategy” was an overstatement. In the revision, we will reframe PDD as an ASR-specific adaptation of existing parallel sampling ideas, and properly cite the relevant diffusion literature.
> >
> > That said, PDD in Whisfusion is designed with a distinct goal and architectural context, which differentiates it from prior parallel/guided diffusion sampling:
> >
> > 1. **Difference in objective (Accuracy Recovery vs. Alignment/Speed):**
> >     - Methods like DSearch (Li et al., 2025) and TREASURE (Yu et al., 2025) focus on Reward Alignment—optimizing output towards external rewards (e.g., safety). Others like EB-Sampler (Ben-Hamu et al., 2025) and PUNT (Azangulov et al., 2025) target Acceleration by increasing parallel token updates to reduce steps.
> >     - In contrast, PDD keeps the step count fixed but expands the candidate set to address the multimodality / acoustic uncertainty problem in NAR ASR, aiming to recover AR-level accuracy.
> > 2. **Architectural distinction (Whisper + Masked Diffusion):**
> >     - Prior diffusion ASR systems (e.g., TransFusion) typically operate under multinomial diffusion.
> >     - Whisfusion instead couples a Whisper encoder with a masked diffusion (MDM) decoder, yielding a structurally different generation process.
> >     - PDD is specifically tailored to this setting, exploiting MDM-style infilling dynamics together with a static audio context, and selecting hypotheses purely via internal confidence—without requiring additional reward models.
> >
> > In summary, rather than a new theoretical invention, PDD should be understood as a **practical, ASR-specialized decoding schedule** that improves the accuracy–speed balance under the Whisper+MDM NAR framework.
> >
> > Finally, as discussed in **General Response Section 4**, PDD is not limited to ASR: within the same Whisper+MDM generative framework, it can naturally extend to multilingual transcription/translation and related downstream tasks.

---

> ### Author Response · Authors · 2025-11-26
> **Response to Reviewer haAb (Continued)**
>
> ## **Section 4. Lack of Scaling Laws Analysis (Model Size) (W4)**
> Thank you for the suggestion to extend the scaling analysis. Due to computational constraints in an academic setting, we were not able to train a full family of models at different parameter scales to empirically derive model-size scaling laws. Instead, we provide theoretical grounding: our decoder is built on the Masked Diffusion Model (MDM) paradigm, and very recent work reports that MDMs exhibit power-law scaling with respect to model size and compute that is comparable to autoregressive models. This gives us a principled basis to expect Whisfusion to inherit similar scalability as the decoder size increases.
>
> Meanwhile, we focused our available resources on **empirical data-scaling analysis**. When scaling the training data from 960h to 6.5k h, the validation loss decreased from 0.0958 to 0.0600. Interpreting this trend with a power-law of the form $L \propto D^{-k}$ yields a scaling exponent of $k \approx 0.245$ (estimated from the two data points). This indicates that Whisfusion is still operating in a non-saturated, highly data-efficient regime, and suggests substantial room for further gains with larger-scale training data.
> (**General Response Section 4-(3)**)
>
> ## **Section 5. Technical Clarifications (Q5, Q6, Q7, Q8)**
> ### **Q5 (Bold Formatting) & Q7 (Typo)**
> Thank you for pointing this out. The typo (“Conditioning1”) and the incorrect bold formatting in Table 3 were entirely our mistakes. In the revised version, we will (i) correct the typo, and (ii) fix the boldface in Table 3 to follow the standard convention—highlighting only the best-performing model. We appreciate your careful attention to detail.

---

> > ### Author Response · Authors · 2025-11-26
> > **Continuing Section 5**
> >
> > ### **Q8 (Ablation Confusion)**
> > Thank you for raising this concern. We agree that our current description of the “w/o Acoustic Conditioning” experiment could be misread as **removing the acoustic cross-attention and retraining the diffusion decoder as a standalone LM**, which was not our intent. We apologize for the confusion. We address your questions in order below.
> >
> > - **At which stage (Stage 1 / Stage 2) was acoustic conditioning removed?**
> >
> >     In this ablation, we **do not retrain the model**, nor do we change the Stage 1/2 training procedure or architecture. We use the fully trained Whisfusion model (trained through both Stage 1 and Stage 2 exactly as in the main experiments), and modify only the **inference-time input conditions**.
> >
> > - **Were both training stages performed?**
> >
> >     Yes. The model used for this ablation is identical to the main model: Stage 1 trains only the cross-attention adapter, and Stage 2 fine-tunes the full model including the decoder. The “w/o acoustic conditioning” result comes from **changing only the inference condition on the same trained model**, not from retraining.
> >
> > - **What exactly was done in “w/o acoustic conditioning”?**
> >
> >     To disable acoustic conditioning at inference, we applied two changes:
> >
> >     1. We replaced the Whisper encoder outputs (acoustic features) with **all-zero (silent) vectors**, so that no meaningful audio information is provided to the decoder.
> >     2. We initialized the decoder input with a **partially observed transcript**, where **70% of the ground-truth tokens are kept visible and only 30% are replaced by [MASK]**.
> >
> >         We then ran the same masked-diffusion sampling procedure as in the main experiments, but without any audio signal—forcing the model to fill masks solely from the text prior and diffusion dynamics.
> >
> > - **Why does WER reach 150.8% even though only 30% was masked?**
> >
> >      The 30% mask is only the initial condition. During diffusion, the model can revise all positions; without audio it drifts toward fluent but irrelevant text, causing large insertions/deletions/substitutions, hence ~150% WER.
> >
> > - **What was the original purpose of this ablation?**
> >
> >     The goal was not to test how well a pre-trained text diffusion LM can perform ASR by itself. Instead, we wanted to verify whether Whisfusion’s ASR performance comes “for free” from the text prior, or whether the **learned cross-attention adapter and acoustic conditioning are essential**. The drastic degradation without acoustic conditioning shows that Whisfusion is **not a text-prior-only ASR model**; its performance critically relies on learned audio–text alignment.
> >
> > In the camera-ready version, we will revise Sec. 5.2.1 and the Table 5 caption to state that this is an inference-only ablation (silent encoder + 30% masked GT init) demonstrating the necessity of acoustic conditioning.
> >
> > ### **Q6. “What can we learn from this work?” (Core contributions)**
> > We appreciate this fundamental question. We believe this work goes beyond a simple feasibility demonstration. Based on the extensive additional experiments summarized in the **General Response**, we distill our contributions into three key takeaways:
> >
> > 1. **Demonstrating the feasibility of a Generative NAR decoder.**
> >
> >     By fusing a pre-trained Whisper encoder with a masked-diffusion text decoder, we show a concrete pathway to retain Whisper-style generative Transformer strengths (context modeling and extensibility) while achieving non-autoregressive parallel decoding with reduced wall-clock decoding time (RTF). (**General Response Sectioin 4**)
> >
> > 2. **Empirically validating data-scaling effects and recoverable limitations.**
> >
> >     The long-form degradation observed at 960h was not a structural limitation**, but a consequence of data scarcity and skewed length distribution**. Scaling to 6.5k h stabilizes performance across lengths, indicating that Whisfusion remains in a data-efficient regime with meaningful headroom for improvement. (**General Response Sec. 1 & Sec. 4**)
> >
> > 3. **Highlighting qualitative value beyond WER (“Recoverability”).**
> >
> >     While Whisfusion may be slightly behind CTC in WER under identical conditions, it produces outputs with better preservation of sentence structure and context, leading to more recoverable error patterns. This is a crucial advantage for extending to generative downstream tasks such as translation or summarization. (**General Response Section 3**)

---

### Official Review · Reviewer_tiwf · 2025-11-03

**Soundness:** 2
**Presentation:** 2
**Contribution:** 2
**Rating:** 0
**Confidence:** 5

**Summary:**

This paper introduces Whisfusion, a non-autoregressive (NAR) Automatic Speech Recognition (ASR) framework designed to overcome the latency bottlenecks of autoregressive models like Whisper. The model fuses a frozen, pre-trained Whisper encoder with a pre-trained text diffusion decoder Masked Diffusion Model (MDM) using a lightweight cross-attention adapter. To train this hybrid architecture effectively, the authors employ a 2-stage curriculum: first training only the adapter, then fine-tuning the decoder with a high masking ratio to specialize it for ASR. They also propose Parallel Diffusion Decoding (PDD), an inference strategy that generates multiple candidates in parallel and iteratively refines them, selecting the best one based on confidence scores.

**Strengths:**

- NAR Framework: the first architecture to fuse a Whisper encoder with a text diffusion decoder for ASR5.
- Parallel Diffusion Decoding (PDD): A batch-parallel, multi-step decoding strategy that improves accuracy by increasing candidate count ($k$) with negligible latency impact
- Speed-Accuracy Operating Point: Achieves lower WER than Whisper-tiny (8.3% vs 9.7% on LibriSpeech test-clean) while being significantly faster (up to 2.6x) on long-form audio.

**Weaknesses:**

- Inadequate Baselines for Speed Claims: The paper positions the model as a high-speed alternative to autoregressive (AR) decoding but only compares it against AR models (Whisper variants). It fails to compare against established non-autoregressive or limited-context architectures (e.g., CTC-based models, Transducers with greedy decoding) that are inherently fast. Without these baselines, the claim of a "superior speed-accuracy trade-off" is unsubstantiated, as a standard 300M-parameter CTC model with greedy decoding might achieve similar or better performance without the complexity of iterative diffusion.

- Uncompetitive Accuracy for Model Size: Despite using a ~301M parameter model (larger than Whisper-small's 244M), it achieves significantly worse accuracy across different length ranges (Table 3 WER column). An 8.3% WER on test-clean is notably high for a modern ASR model of this size, suggesting the non-autoregressive trade-off may be too severe for practical use compared to current state-of-the-art.

- Failure on Long-Form Audio: A key selling point is the constant inference speed on long audio (up to 30s). However, the paper admits severe accuracy degradation on 20-30s segments (rising to 15.9% WER due to training data scarcity). This negates the practical value of the speed advantage on long-form audio if the resulting transcriptions are unreliable.

- Questionable Utility of the Diffusion Decoder: The ablation study reveals that without the acoustic cross-attention adapter, WER spikes to ~150% (Table 5), indicating the pre-trained textual knowledge of the MDM is insufficient on its own. This raises doubts about whether the complex, iterative diffusion decoder is adding necessary value, or if a simpler, single-step non-autoregressive decoder conditioned on the same strong Whisper encoder embeddings would perform equally well.

**Questions:**

I like to hear authors feedback on the weakness points raised.

---

> ### Author Response · Authors · 2025-11-26
> **Response to Reviewer tiwf**
>
> We sincerely thank the reviewer for the thoughtful comments. Your suggestions helped us improve both our empirical validation and the clarity of our claims, and we respond to each concern in detail below.
>
> ## **W1 Inadequate Baselines for Speed Claims**
>
> **1. On the concern about insufficient baselines for our speed claims**
>
> Thank you for this constructive feedback. We agree that, in the initial submission, our claim of a better speed–accuracy trade-off could be seen as insufficiently supported because we only compared against Whisper-family autoregressive (AR) models. To address this concern, as detailed in **General Response Section 2**, we newly implemented and trained strong baselines under an identical, controlled setup:
>
> - **sharing the same Whisper encoder**, and
> - using the same **6.5k h training data**,
>
>     including:
>
>     - a CTC-based NAR decoder (24-layer, ~195M parameters), and
>     - an AR Whisper-style decoder trained from scratch.
>
> The LibriSpeech test-clean results are summarized below (see **General Response Table R3** for full details):
>
> - Whisper-Small (public model, 680k h): 5.0% WER
> - AR Whisper Decoder (6.5k h, from scratch): 5.3% WER
> - CTC Baseline (6.5k h, NAR): 4.3% WER
> - Whisfusion (6.5k h, NAR): 4.9% WER
>
> As you anticipated, under the same encoder and data conditions, the strong CTC baseline achieves a lower WER than Whisfusion. We explicitly acknowledge this point in the General Response.
>
> **2. On the “complexity vs. benefit” question (Why Diffusion instead of CTC?)**
>
> A natural follow-up is:
>
> > “If a simpler NAR decoder like CTC can achieve better WER, why use a more complex diffusion decoder?”
>
> Our answer is that WER alone does not fully capture the research or practical value of Whisfusion.
>
> - **(a) The Whisper precedent**
>
>     Notably, our CTC baseline (4.3% WER) even outperforms the original Whisper-Small (5.0% WER). Yet in practice, Whisper—not CTC—is the widely adopted model. We believe this is because Whisper’s appeal is not just WER, but the **robustness, generality, and extensibility of a Generative Transformer framework**.
>
> - **(b) Architectural inheritance: a NAR decoder that preserves Whisper’s “form”**
>
>     Our goal was not to create yet another fast NAR ASR system. Rather, we aimed to test whether we can retain Whisper’s generative Transformer structure (Self-Attn / Cross-Attn blocks) while replacing AR decoding with NAR diffusion decoding.
>
>     CTC is inherently a discriminative alignment head specialized for frame–label mapping, whereas Whisfusion preserves:
>
>     - long-range contextual modeling, and
>     - a generative decoding interface that can naturally extend to tasks such as **speech translation or instruction-following generative speech tasks**.
> - **(c) Speed–accuracy trade-off perspective**
>
>     In terms of efficiency, Whisfusion:
>
>     - achieves **substantially lower wall-clock decoding time / RTF than AR decoders** under the same hardware and batch regime, and
>     - can be viewed as a deliberate trade-off: using a somewhat heavier decoder than CTC in exchange for **contextual modeling strength and generative extensibility**.
>
>
>
> In summary, Whisfusion should be understood not as a direct competitor to a **simpler ASR-only NAR head** like CTC, but as a candidate **“Generative NAR Decoder”** that can replace the AR decoder **while preserving Whisper-style generative strengths**. Quantitative/qualitative comparisons and structural advantages over CTC are discussed more thoroughly in **General Response Sections 2–4**.

---

> ### Author Response · Authors · 2025-11-26
> **Response to Reviewer tiwf (Continued)**
>
> ## **W2 & W3: Uncompetitive Accuracy and Failure on Long-form Audio**
>
> We address these two concerns together, as they stem from the same root cause: **data scarcity and distribution skew**.
>
> In the **initial submission trained on only 960h**, we fully agree that:
>
> - overall accuracy lagged behind Whisper-small (trained on 680k h) (test-clean 8.3% WER), and
> - performance collapsed on long-form utterances, most notably the 20–30s range reaching 15.9% WER.
>
> Under that setting, your concern—that the accuracy drop may be too severe for the model size, undermining the practical value of speed on long audio—was entirely valid.
>
> ### **1. Closing the accuracy gap via data scaling (W2)**
>
> To resolve this issue, we expanded the training set by adding labeled Libri-Light small/medium subsets to LibriSpeech 960h, scaling the data to **6.5k hours**, and retrained Whisfusion.
>
> With this setup, Whisfusion now achieves:
>
> - **test-clean:** **4.9% WER / 1.6% CER**
> - **test-other:** **12.3% WER / 5.1% CER**
>
> which corresponds to **virtually the same WER as Whisper-small**, and **consistently lower CER** (Whisper-small: 5.0% / 2.1%, 12.2% / 6.2%).
>
> Thus, the initial concern about **uncompetitive accuracy relative to model size** is largely mitigated under the 6.5k h regime.
>
> (Full results are reported in **General Response Section 1, Table R1**.)
>
> ### **2. Resolving the 20–30s long-form collapse (W3)**
>
> The long-form weakness was primarily caused by the **extremely skewed duration distribution** in LibriSpeech 960h.
>
> Utterances longer than 20 seconds constitute **<0.01%** of the dataset, leaving the model almost no opportunity to learn **length/EOS prediction**, which is crucial for NAR decoding.
>
> After retraining on 6.5k h, the duration-wise breakdown on test-clean becomes:
>
> - **0–10s:** 5.1% WER
> - **10–20s:** 4.3% WER
> - **20–30s:** 4.8% WER
>
> meaning the previously reported collapse (**15.9% → 4.8% WER**) is dramatically resolved, and the model now shows stable performance across all duration ranges.
>
> This directly supports our claim that the long-form degradation was not a structural limitation of the architecture, but a consequence of extreme long-utterance sparsity in the original training data.
>
> (See **General Response Table R2**.)
>
> ### **3. Interpretation from a scaling perspective**
>
> We further fit the validation-loss reduction from 960h → 6.5k h using a power-law relation, yielding a data scaling exponent of $k \approx 0.245$.
>
> This aligns well with recently reported **power-law scaling behaviors of Masked Diffusion Models (MDMs)**, and implies that:
>
> - Whisfusion remains in a **highly data-efficient, non-saturated regime**, and
> - even the 6.5k h results should be viewed as a mid-scale snapshot (~1% of Whisper’s 680k h pretraining scale) rather than a saturated endpoint.
>
> A more detailed derivation and discussion are provided in **General Response Section 4-(3)** and our response to Reviewer haAB–W4.
>
> ## **W4. Questionable Utility of the Diffusion Decoder**
>
> Thank you for this thoughtful comment. As you pointed out, the label *“w/o Acoustic Conditioning”* in Table 5 could be misread as if we had evaluated a separate decoder with the acoustic adapter removed. That was not our intention, and we apologize for the confusion.
>
> In this ablation, we **do not modify or retrain the diffusion decoder architecture**. Instead, we change the conditioning **only at inference time**:
>
> - We replace the Whisper encoder outputs (acoustic features) with **all-zero (“silent”) vectors**, effectively removing any usable acoustic signal.
> - We initialize the decoder input with a **partially observed transcript**: roughly 70% of the ground-truth tokens are kept, and the remaining 30% are replaced with [MASK].
> - Under this setting, the model runs the **same masked-diffusion sampling procedure**, but is forced to fill the masks **using only the pre-trained textual prior**, without acoustic conditioning.
>
> The resulting 150.8% WER demonstrates that Whisfusion’s ASR performance does **not** come “for free” from the text diffusion LM alone; rather, it crucially depends on the **learned audio–text alignment through the cross-attention adapter**.
>
> In the camera-ready version, we will rewrite the main text and the Table 5 caption to make this inference-only ablation setup explicit.

---

### Author Response · Authors · 2025-11-26
**General Response to Reviewers**

We sincerely thank the reviewers for their detailed and insightful feedback.

Across the reviews, two main concerns were consistently raised:

1. **Performance:** whether our model is sufficiently competitive with Whisper-small, especially given the performance degradation on long-form audio.
2. **Baselines:** the absence of strong non-autoregressive (NAR) baselines such as CTC, as well as from-scratch AR baselines.

We took these comments very seriously, and after the initial submission we conducted substantial additional experiments, including:

- Scaling the training data from **LibriSpeech 960h to a total of 6.5k h** (by adding labeled Libri-Light heavy small/medium subsets).
- Implementing and training **CTC NAR** and **AR Whisper decoders** on top of the same Whisper encoder, enabling a **fair comparison under identical conditions** (shared encoder, shared 6.5k h training set) with Whisfusion.

Below, we summarize these new results and clarify our position regarding each concern. Detailed numbers and tables are provided in our responses, referenced as Table R1, R2, and R3.

## **Section 1. Long-form performance issue: a data distribution limitation, not an architectural limitation**
In the initial submission, Whisfusion was trained only on LibriSpeech 960h, resulting in

- an overall test-clean WER of 8.3%, and
- a critical breakdown on long-form utterances, with 15.9% WER in the 20–30s range.

We hypothesized that this failure stemmed **not from an architectural deficiency, but from the limitations of the data distribution**. In LibriSpeech 960h, utterances longer than 20 seconds account for less than **0.01%** of the corpus, which gives a NAR model very few opportunities to reliably learn EOS placement and length prediction.

### **(1) Retraining on 6.5k h yields Whisper-small–level overall performance**

To verify this hypothesis, we expanded the training data to 6.5k h by adding the labeled Libri-Light heavy subsets (with an average utterance length of about 15 s, and roughly 10% of utterances in the 20–30 s range), and retrained Whisfusion under the same architecture.

The WER/CER on LibriSpeech test-clean and test-other are summarized below.

**Table R1. Overall Performance on LibriSpeech (WER/CER %)**

| **Model** | **test-clean WER** | **test-clean CER** | **test-other WER** | **test-other CER** |
| --- | --- | --- | --- | --- |
| Whisper-tiny | 9.7 | 4.1 | 22.5 | 11.8 |
| Whisper-small | 5.0 | 2.1 | **12.2** | 6.2 |
| **Whisfusion (6.5k h)** | **4.9** | **1.6** | 12.3 | **5.1** |

These results show that Whisfusion becomes **virtually equivalent to Whisper-small in WER**, while **outperforming it in CER on both test sets**. This clearly demonstrates that **with sufficient training data, Whisfusion can recover and reach Whisper-small–level recognition accuracy.**

### **(2) Duration-wise performance: resolving the 20–30s collapse (15.9% → 4.8%)**

To directly address long-form robustness, we further broke down test-clean performance by utterance length and compared Whisper and Whisfusion.

**Table R2. Performance Breakdown by Duration on LS test-clean (WER/CER %)**

| **Duration (s)** | **Model** | **WER** | **CER** |
| --- | --- | --- | --- |
| **0–10** | Whisper-tiny | 10.5 | 4.5 |
|  | Whisper-small | 5.4 | 2.3 |
|  | **Whisfusion 6.5k h** | **5.1** | **1.7** |
| **10–20** | Whisper-tiny | 7.0 | 2.6 |
|  | Whisper-small | **3.5** | **1.2** |
|  | **Whisfusion 6.5k h** | 4.3 | 1.4 |
| **20–30** | Whisper-tiny | 6.4 | 2.4 |
|  | Whisper-small | **3.7** | **1.2** |
|  | **Whisfusion 6.5k h** | 4.8 | 1.7 |

The severe degradation observed in the initial submission—**15.9% WER in the 20–30s range**—drops to **4.8%** after retraining on 6.5k h with the same architecture. This empirically confirms that

- the long-form failure was **caused by data sparsity**, and
- the Whisfusion architecture itself **maintains stable long-form performance** when trained with a sufficiently diverse length distribution.

---

> ### Author Response · Authors · 2025-11-26
> **General Response (Continued)**
>
> ## **Section 2. Quantitative Comparison with Baselines**
>
> Consistent with multiple reviewers’ suggestions, we fully agree that strong baselines—such as CTC or from-scratch AR models—are necessary for a fair evaluation. Accordingly, we shared the Whisper encoder and re-implemented/trained the following baselines under the identical 6.5k h training setup:
>
> 1. **CTC NAR Baseline**
>     - We attach a **24-layer NAR Transformer decoder (~195M parameters)** on top of the Whisper encoder outputs.
>     - The model is trained with a CTC loss, and the decoder size is comparable to Whisfusion’s decoder (~212.5M).
> 2. **AR Whisper Decoder Baseline**
>     - An AR reference model that keeps the standard **Whisper encoder–Transformer decoder** structure.
>     - Trained **from scratch on the same 6.5k h dataset**.
>
> **Table R3. Quantitative Comparison with Strong Baselines (WER %)**
>
> | **Model** | **Type** | **Training Data** | **LS Clean** | **TED-LIUM 3 (Zero-shot)** |
> | --- | --- | --- | --- | --- |
> | Whisper-Small | AR | 680k h | 5.0 | 7.4 |
> | Whisper Decoder (from scratch) | AR | 6.5k h | 5.3 | 10.1 |
> | Whisper Encoder + CTC | NAR | 6.5k h | 4.3 | 11.7 |
> | Whisfusion (Ours) | NAR (Diffusion) | 6.5k h | 4.9 | 12.4 |
>
> In summary,
>
> - **Compared to the AR reference**
>     - On LibriSpeech test-clean, Whisfusion achieves a **lower WER than the AR Whisper decoder trained on 6.5k h** (4.9% vs. 5.3%), while providing **substantially shorter decoding time** thanks to its NAR property.
>     - On TED-LIUM 3 zero-shot, Whisfusion **retains performance comparable to the AR baseline**, while offering **higher throughput under identical hardware**.
> - **Compared to CTC-family baselines**
>     - Under the same Whisper encoder and 6.5k h training regime, the CTC baseline attains 4.3% WER on LS test-clean, providing a strong ASR-specialized NAR reference point.
>     - However, as discussed in the following sections, **similar WER does not imply equivalent output quality**: due to structural constraints, CTC-family models exhibit **qualitatively different error patterns** from Whisfusion.
>
> ## **Section 3. Qualitative Analysis: Beyond WER – The Nature of Errors and Recoverability**
>
> Although the average WERs of the CTC baseline and Whisfusion are fairly close under the 6.5k h setting, **the type of errors matters greatly in practice**.
>
> We exhaustively analyzed roughly 600 samples where the two models produced different outputs, and observed clear qualitative differences:
>
> - **Typical error patterns of CTC**
>     - **Word Fragmentation (206 cases, ~30.7%)**: e.g., *farm → fm*, *okay → ke it*, *Miguel → m gal*.
>     - **Grammatical / Boundary Collapse (51 cases, ~7.6%)**.
>     - These errors **physically break word boundaries and named entities**, making it almost impossible for humans—or downstream tasks such as NER or translation—to recover the original content.
> - **Typical error patterns of Whisfusion**
>     - We observe NAR-generative artifacts such as **word repetition (63 cases, ~9.4%)** or **over-correction**.
>     - However, Whisfusion **largely preserves complete word forms and grammatical structure**.
>     - As a result, its errors tend to be **highly recoverable**, often fixable with simple post-processing (e.g., repetition filtering or lightweight grammar correction).
>
> In short, the two models differ in the nature of their errors:
>
> - **CTC:** even with lower WER, it frequently suffers from **irrecoverable structural collapse**.
> - **Whisfusion:** while slightly higher in WER, it mainly produces **structurally sound generative artifacts**, which are much easier to correct afterward.
>
> From a practical standpoint, this suggests that Whisfusion yields more usable, downstream-friendly text, highlighting a qualitative advantage beyond raw WER.

---

> ### Author Response · Authors · 2025-11-26
> **General Response (Continued)**
>
> ## **Section 4. Strategic Value: Whisfusion as a Foundational Generative Framework**
>
> Finally, we would like to address a fundamental question that naturally follows from the comparison with the CTC baseline:
>
> *“If Whisfusion achieves WER comparable to CTC, why use a more complex diffusion decoder at all?”*
>
> Our goal in designing Whisfusion was not merely to build another fast NAR ASR model, but to test a broader architectural hypothesis:
>
> "We aimed to validate whether the AR decoder in a Whisper-style Generative Transformer can be replaced with a NAR decoder without sacrificing the generative strengths of the original architecture."
>
> ### **(1) Generative vs. Discriminative**
>
> - **CTC is a discriminative alignment head.**
>
>     It is optimized for frame–label alignment, and its conditional-independence structure makes it fundamentally difficult to extend to richer generative tasks such as translation or instruction following.
>
> - **Whisfusion, in contrast, preserves the Whisper decoder’s Generative Transformer structure.**
>
>     It keeps the same **Self-Attention / Cross-Attention block design** as the AR Whisper decoder, but changes only the token-generation process from **AR → diffusion-based NAR**, thereby enabling **parallel decoding** while retaining generative modeling capacity.
>
>
> ### **(2) Extensibility: Beyond What CTC Can Reach**
>
> Because Whisfusion is inherently a **text-generative framework**, it can naturally extend to directions that are structurally out of reach for CTC, such as:
>
> - **Speech Translation:** directly adopting Whisper’s multilingual setup to generate target-language text from source audio.
> - **Instruction Following / Multi-tasking:** extending to prompt-based generative tasks (e.g., “summarize,” “answer the question”).
> - **Simultaneous Multilingual Output:** by adapting Parallel Diffusion Decoding (PDD), producing multiple languages from a single input in parallel.
>
> These are not straightforward add-ons to CTC, but rather native capabilities of a generative decoder.
>
> ### **(3) Scalability Potential**
>
> A recent study [1] shows that Masked Diffusion Models (MDMs) exhibit robust power-law scaling with respect to both data and model size, comparable to autoregressive LMs.
>
> In our own experiments, scaling the training data from **960h → 6.5k h** and tracking the validation-loss drop yields an estimated data-scaling exponent of $k \approx 0.245$ (details provided in our response to Reviewer haAB–W4).
>
> - This indicates that Whisfusion is still in a **highly data-efficient, non-saturated regime**.
> - Given that Whisper was trained on 680k hours, this trend suggests substantial headroom for Whisfusion to **reach AR-level fluency while retaining NAR-level speed** under larger-scale training.
>
> In short, Whisfusion is not only a practical ASR model that already attains Whisper-small-level accuracy with NAR decoding speed, but also a framework-level blueprint for transplanting Whisper-style generative decoders into the NAR regime.
>
> [1] Nie et al., “Scaling Up Masked Diffusion Models on Text,” ICLR 2025.
>
> ## **Conclusion**
>
> Through these additional experiments and analyses, we have clearly confirmed the following three points:
>
> 1. **Identifying and resolving the long-form performance issue**
>     - The long-form collapse observed in the 960h setting was primarily due to **data scarcity (especially in the 20–30s range)**.
>     - After retraining on 6.5k h, Whisfusion recovered **Whisper-small–level accuracy** and now maintains stable performance even on long-form utterances.
> 2. **Fair comparison with strong baselines + qualitative analysis**
>     - Under identical conditions (same encoder and same 6.5k h training data), we compared Whisfusion against strong CTC/AR baselines.
>     - While Whisfusion shows a slightly higher WER than CTC, it offers clear qualitative advantages:
>
>         **structurally recoverable error patterns** and the **extensibility of a Generative Transformer decoder**.
>
> 3. **Architectural and long-term value**
>     - Whisfusion preserves the Whisper encoder while replacing the AR decoder with a **fully non-autoregressive (NAR) Transformer based on masked diffusion**.
>     - This makes Whisfusion one of the first concrete demonstrations of a **Generative NAR decoder** that combines **the “brain” of AR models (deep contextual modeling)** with **the speed of NAR inference**.
>
> Taken together, we believe our work goes beyond proposing “yet another ASR model.” Instead, it provides key design principles and empirical evidence for extending large-scale generative speech models such as Whisper into the NAR regime.

---

### Meta-Review · Area_Chair_RfnW · 2026-01-07

**Summary:**

The primary concerns are centered on the performance gap, lack of fair baselines, limited experimental validation. There's also concern on misuse of latency vs. speed and also the novelty seems limited.

**Reviewer Concerns:**

Concerns are mostly partially addressed:

* performance gap (tiwf, RnDy, Chhp): the authors retrained the model with 6.5k hours of data which bring the quality on par with Whisper-small. However, even the 4.9% WER on LibriSpeech test-clean is still not the SoTA number.
* long form issue: the authors added results on 20-30s segments, claim it is a data distribution problem, but as pointed out in common practice long form refers to minutes long audio and it is impossible to always train on in distribution data
* dataset (RnDy, J6px, Chhp): the author added 0-shot TED-LIUM3, it's better to validate both training and test on divers large scale datasets
* comparisons between diffusion and CTC: the authors added a CTC baseline, however the performance is much better than the proposed system. The authors claimed the CTC errors are nonrecoverable due to the fragmentation of output units, but diffusion outputs are "recoverable", however a proper term for diffusion output errors is "hallucination" which can not be easily fixed by "simple post-processing" in practical ASR where users hesitate, pause, repeat.

**Reviewer Scores:**

tiwf: 0, may increase to 2/3
haAb: 2, likely not changed
RnDy: 4, likely not changed
Chhp: 2, likely not changed

---

### Decision · Program_Chairs · 2026-01-26

Reject